# Compile Scene Graphs with Reinforcement Learning

## Abstract

Next token prediction is the fundamental principle for training large language models (LLMs), and reinforcement learning (RL) further enhances their reasoning performance. As an effective way to model language, image, video, and other modalities, the use of LLMs for end-to-end extraction of structured visual representations, such as scene graphs, remains underexplored. It requires the model to accurately produce a set of objects and relationship triplets, rather than generating text token by token. To achieve this, we introduce *R1-SGG*, a multimodal LLM (M-LLM) initially trained via supervised fine-tuning (SFT) on the scene graph dataset and subsequently refined using reinforcement learning to enhance its ability to generate scene graphs in an end-to-end manner. The SFT follows a conventional prompt-response paradigm, while RL requires the design of effective reward signals. We design a set of graph centric rewards, including three recall based variants—Hard Recall, Hard Recall+Relax, and Soft Recall—which evaluate semantic and spatial alignment between predictions and ground truth at the object and relation levels. A format consistency reward further ensures that outputs follow the expected structural schema. Extensive experiments on the VG150 and PSG benchmarks show that R1-SGG substantially reduces failure rates and achieves strong performance in Recall and mean Recall, surpassing traditional SGG models and existing multimodal language models.

## 1   Introduction

Scene graphs, as structured visual representations, have gained increasing attention in many vision applications, such as robot manipulation [44, 41], robot navigation [7, 23, 37], and medical image or video analysis [20, 24], *etc*. To generate scene graphs from an image, traditional Scene Graph Generation (SGG) models [10, 34, 14, 38, 29, 2, 15, 11, 5, 40, 4] decouple the task into two subtasks, *i.e.*, object detection and visual relationship recognition, and directly maximize the likelihood of the ground-truth labels given the image. Essentially, these models tend to overfit the distribution of annotated datasets; Consequently, they struggle to handle long-tail distributions and are prone to generating biased scene graphs (*e.g.*, all predicted relationships are head classes like "on" and "of").

While traditional SGG models rely on manual annotated datasets and struggle to generalize to new domains, recent advances in large langue models (LLMs) offer a new paragdim. LLM4SGG [12] utilizes an LLM to extract relationship triplets from captions using both original and paraphrased text, while GPT4SGG [3] employs an LLM to synthesize scene graphs from dense region captions. Additionally, Li [17] generates scene graphs via image-to-text generation using vision-language models (VLMs). These weakly supervised methods demonstrate potential for generating scene graphs with little or no human annotation but suffer from accuracy issues in the generated results.

Despite these advancements, existing methods typically employ text-only LLMs or rely on intermediate captions as input, which do not fully leverage the rich visual context. In contrast, multimodal large language models (M-LLMs) which integrate both visual and linguistic modalities offer the

Submitted to 39th Conference on Neural Information Processing Systems (NeurIPS 2025). Do not distribute.

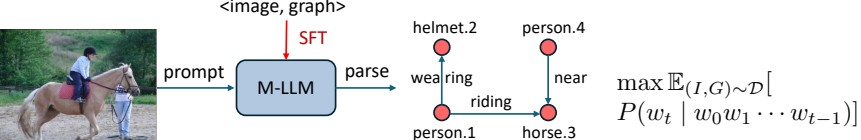

(a) M-LLM with SFT is optimized token by token (here, $w_i$ refers to a token).

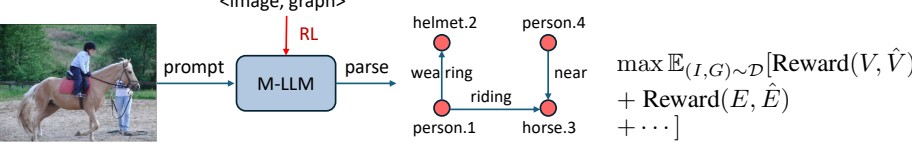

(b) M-LLM with RL is optimized using rule-based rewards. Here, $G = (V, E)$ and $\hat{G} = (\hat{V}, \hat{E})$ refer to the ground-truth and predicted scene graphs, respectively.

Figure 1: Comparison of multimodal LLMs (M-LLMs) fine-tuned via Supervised Fine-tuning (SFT) and Reinforcement Learning (RL) for Scene Graph Generation (SGG).

potential for more direct and holistic scene understanding. By processing visual information alongside natural language prompts, M-LLMs can generate scene graphs in an end-to-end manner. However, in practice, M-LLMs suffer from instruction following (*e.g.*, the output does not contain "objects" or "relationships"), repeated response (*e.g.*, *{"objects":[··· {"id": "desk.9", "bbox": [214, 326, 499, 389]}, {"id": "desk.10", "bbox": [214, 326, 499, 389]}, {"id": "desk.11", "bbox": [214, 326, 499, 389]}, ··· ]··· }* ), inaccurate location, *etc*. These challenges highlight the need for better alignment between visual understanding and structured representation within the M-LLM framework.

To improve instruction-following and structured output generation in M-LLMs, one intuitive solution is to perform Supervised Fine-tuning (SFT) on scene graph datasets (see Fig. 1-(a)). In the context of SGG, SFT aligns the model's outputs with expected formats (*e.g.*, structured lists of objects and relationships) by training it on high-quality scene graph annotations. This process encourages the model not only to recognize entities and relations from the image but also to organize them into a coherent and valid graph structure. Nevertheless, SFT alone still be insufficient as all output tokens are weighted equally in the loss. For example, the experimental results on the VG150 dataset [34] reveal that even with SFT, M-LLM still has a high failure rate to generate a valid and high-quality scene graph. The drawback of SFT in SGG lies in the lack of effective signals to correct the output (*e.g.*, the model cannot directly utilize the Intersection over Union (IoU) between the predicted box and the ground truth to refine its output ).

To advance M-LLMs for effective Scene Graph Generation (SGG), we propose *R1-SGG*, a novel framework leveraging visual instruction tuning enhanced by reinforcement learning (RL). The visual instruction tuning stage follows a conventional supervised fine-tuning (SFT) paradigm, *i.e.*, fine-tuning the model using prompt-response pairs with a cross-entropy loss. For the RL stage, we adopt GRPO, an online policy optimization algorithm introduced in DeepSeekMath [28].

To enable effective reinforcement learning for Scene Graph Generation, we introduce a set of rule-based, graph-centric rewards that reflect the structural characteristics of scene graphs. Given an image and a prompt, a multimodal large language model (M-LLM) generates a set of objects and relational triplets. To evaluate and optimize these predictions, we formulate reward functions aligned with standard SGDET metrics [34] and structured reasoning objectives. Specifically, we define three reward variants: **Hard Recall**, which counts a triplet as correct only if the subject, predicate, and object labels exactly match the ground truth and both bounding boxes achieve IoU > 0.5; **Hard Recall+Relax**, which relaxes the exact match constraint by incorporating embedding similarity between predicted and ground-truth labels; and **Soft Recall**, which further densifies reward signals via bipartite matching, combining object label similarity, IoU, and bounding box distance into a unified cost function. These scene graph rewards are computed over matched object and edge pairs, and are complemented by a format reward that enforces structural adherence in output formatting.

This reward design enables stable and fine-grained policy optimization using GRPO, guiding the M-LLM toward generating accurate, complete, and structurally valid scene graphs.

Our contributions can be summarized as follows:

- We explore how to develop a multimodal LLM for Scene Graph Generation (SGG), by leveraging visual instruction tuning with reinforcement learning (RL). To our knowledge, this is a pioneer work that develop a multimodal LLM to generate scene graphs in an end-to-end manner.

- Graph-centric, rule-based rewards are designed to guide policy optimization in a manner aligned with standard evaluation metrics in SGG, such as the recall of relationship triplets—metrics that cannot be directly optimized through SFT.

- Experimental results demonstrate that the proposed framework improves the ability to understand and reason about scene graphs for multimodal LLMs.

## 2 Related Work

**Scene Graph Generation (SGG).** Scene Graph Generation (SGG) is a foundational task in structured visual understanding, where the goal is to represent an image as a graph of objects and their pairwise relationships. Traditional approaches like [34, 14, 38, 29, 2, 15, 11, 5] decouple the task into object detection and relationship classification stages, and are typically trained via supervised learning on datasets such as Visual Genome (VG150) [34]. While effective, these models are limited by their reliance on annotated data and exhibit strong bias toward head predicates such as "on" or "of", struggling on long-tail classes.

To overcome the closed-set limitation, recent work has explored open-vocabulary SGG. For example, OvSGTR [4] extends scene graph prediction to a fully open-vocabulary setting by leveraging visual-concept alignment. In parallel, weakly supervised methods have been developed to reduce the annotation burden. These approaches, such as those proposed by [43, 16, 40, 4], use image-caption pairs as supervision to distill relational knowledge, enabling generalization to unseen concepts.

**LLMs for Scene Graph Generation.** With the rise of LLMs, several studies have attempted to synthesize scene graphs from natural language. LLM4SGG [12] extracts relational triplets from both original and paraphrased captions using text-only LLMs. GPT4SGG [3] goes a step further by using GPT-4 to generate scene graphs from dense region captions, improving contextual consistency and coverage. Meanwhile, [17] leverage vision-language models (VLMs) to produce scene graphs through image-to-text generation pipelines.

However, these caption-based or LLM-driven methods often exhibit limited accuracy, including incomplete object sets, and inconsistent relationship descriptions. These issues arise from the lack of structure in the generated outputs and the absence of mechanisms to refine the results according to scene-level constraints.

**Reinforcement Learning (RL) for LLMs.** Reinforcement learning (RL) has been increasingly adopted to enhance the reasoning capabilities of large models. Algorithms like Proximal Policy Optimization (PPO) [27] and Group Relative Policy Optimization (GRPO) [28] guide models using reward signals instead of relying solely on maximum likelihood estimation. In the context of large language models, DeepSeek-R1 [8] demonstrates that RL can significantly improve structured reasoning and planning.

In multimodal learning, however, RL remains underutilized for generating structured outputs. Our work addresses this by introducing rule-based reward functions at multiple levels, including three scene graph reward variants and a format consistency reward. These signals promote the generation of meaningful and coherent scene graphs by explicitly evaluating alignment with ground-truth annotations.

## 3 Methodology

### 3.1 Preliminary

**Scene Graph Generation (SGG).** Scene graph generation (SGG) transforms an image $I$ into a structured representation that captures both objects and their interactions. Specifically, SGG produces a directed graph $\mathcal{G} = (\mathcal{V}, \mathcal{E})$, where each node $v_i \in \mathcal{V}$ represents an object annotated with an object category $c_i$ and a bounding box $b_i$. Each relationship triplet $e_{ij} \in \mathcal{E}$ captures the relationship between two nodes. The triplet is defined as $e_{ij} := \langle v_i, p_{ij}, v_j \rangle$, where $p_{ij}$ encodes the visual relationship between the subject $v_i$ and the object $v_j$, such as spatial relations (*e.g.*, "on", "under") or interactive relations (*e.g.*, "riding", "holding"). Typically, SGG models decouple this task into two subtasks, namely object detection and relationship recognition, both optimized by maximizing the likelihood of the corresponding ground-truth labels given the image.

**Reinforcement Learning with GRPO.** Group Relative Policy Optimization (GRPO) is a online reinforcement learning algorithm introduced by DeepSeekMath [28]. Unlike traditional methods such as PPO [27], which require an explicit critic network, GRPO instead compares groups of candidates to update the policy $\pi_\theta$. Specifically, for each input query $q$, a set of candidate outputs $\{o_i\}_{i=1}^{G}$ is drawn from the previous policy $\pi^{\text{old}}(O|q)$, and the advantage of each candidate is computed relative to the group's average reward:

$$A_i = \frac{r_i - \text{mean}(\{r_1, \ldots, r_G\})}{\text{std}(\{r_1, \ldots, r_G\})}. \tag{1}$$

The policy parameters $\theta$ are updated by maximizing the following GRPO objective:

$$
\begin{aligned}
J_{\text{GRPO}}(\theta) = \mathbb{E}_{q \sim P(Q), \{o_i\}_{i=1}^{G} \sim \pi^{\text{old}}(O|q)} &\left[ \frac{1}{G} \sum_{i=1}^{G} \left( \min \left( \frac{\pi_\theta(o_i|q)}{\pi^{\text{old}}(o_i|q)} A_i, \right. \right. \right. \\
&\left. \left. \left. \text{clip}\left( \frac{\pi_\theta(o_i|q)}{\pi^{\text{old}}(o_i|q)}, 1-\epsilon, 1+\epsilon \right) A_i \right) - \beta \, D_{\text{KL}}\big(\pi_\theta \,\|\, \pi_{\text{ref}}\big) \right],
\end{aligned} \tag{2}
$$

Here, $\epsilon$ and $\beta$ are hyper-parameters. The first term uses a clipped probability ratio (as in PPO) to control the update magnitude, while the KL divergence regularizer $D_{\text{KL}}(\pi_\theta \,\|\, \pi_{\text{ref}})$ constrains the new policy $\pi_\theta$ to not deviate too much from a reference policy $\pi_{\text{ref}}$. This formulation, which combines a group-relative advantage, a clipping mechanism, and a KL divergence regularizer, stabilizes policy updates and improves training efficiency, demonstrating remarkable potential for enhancing the reasoning performance of LLMs such as DeepSeek R1 [8].

### 3.2 Overview of R1-SGG

R1-SGG is a reinforcement learning framework that enhances scene graph generation (SGG) in multimodal large language models (M-LLMs). It builds on a supervised fine-tuning (SFT) stage using prompt-response pairs, followed by reinforcement learning (RL) with structured, graph-centric rewards.

Given an input image and prompt, the M-LLM generates a scene graph $\mathcal{G}_{\text{pred}} = (\mathcal{V}_{\text{pred}}, \mathcal{E}_{\text{pred}})$, comprising objects (nodes) and their relationships (edges). We primarily optimize using *Hard Recall*, which aligns with SGDET metrics by rewarding exact triplet matches. To study the sparsity and design of the rewards, we also evaluated relaxed alternatives based on bipartite matching between $\mathcal{G}_{\text{pred}}$ and the ground truth graph $\mathcal{G}_{\text{gt}}$, allowing fine-grained node and edge rewards. Our RL pipeline employs Group Relative Policy Optimization (GRPO) [28], which compares sampled outputs and promotes higher-reward candidates. By integrating SFT, GRPO, and graph-aware rewards, R1-SGG enables M-LLMs to generate accurate, diverse, and structurally valid scene graphs.

### 3.3 Rewards Definition

#### 3.3.1 Format Reward

Following DeepSeek R1 [8], we employ a format reward to ensure that the model's response adheres to the expected structure, specifically `<think>···</think><answer>···</answer>`. A reward of 1

is assigned if the response follows this format and the segment enclosed by `<answer>···</answer>` contains both the keywords `"object"` and `"relationships"`; otherwise, the reward is 0.

### 3.3.2 Scene Graph Rewards

Standard evaluation protocols for Scene Graph Generation (SGG), such as SGDET [34], formulate the task as a recall-oriented problem, emphasizing the model's ability to retrieve correct relationship triplets from an image. To investigate the impact of different reward formulations, we introduce three variants: *Hard Recall*, *Hard Recall+Relax*, and *Soft Recall*.

**Hard Recall.** To align policy optimization with standard SGDET metrics, we define *Hard Recall*, where a predicted triplet $\langle \texttt{subject}, \texttt{predicate}, \texttt{object} \rangle$ is counted as a true positive when *both* of the following hold: 1) *Triplet accuracy:* the subject, predicate, and object labels exactly match the ground truth. 2) *Localization accuracy:* the IoU between predicted and ground-truth bounding boxes exceeds 0.5.

This reward is aligned with standard metrics but suffers from sparsity due to its strict criteria.

**Hard Recall + Relax.** We relax the triplet accuracy requirement by computing cosine similarity between the entity embeddings of predicted and ground-truth triplets. This softens the discrete matching constraint to provide more gradient signal.

**Soft Recall.** We further propose a dense matching reward by formulating it as a bipartite matching problem, similar to DETR [1], where predicted nodes $\{v_i = (c_i, b_i)\}_{i=1}^M$ (each node $v_i$ is comprising an object class $c_i$ and a bounding box $b_i$) are matched to ground-truth nodes $\{\tilde{v}_j = (\tilde{c}_j, \tilde{b}_j)\}_{j=1}^N$ with the following cost:

$$
\begin{aligned}
\text{cost}(v_i, \tilde{v}_j) =& \lambda_1 \cdot (1.0 - \langle \texttt{Embedding}(c_i), \texttt{Embedding}(\tilde{c}_j) \rangle) \\
& + \lambda_2 \cdot (1.0 - \text{IoU}(b_i, \tilde{b}_j)) + \lambda_3 \cdot ||b_i - \tilde{b}_j||_1,
\end{aligned}
\tag{3}
$$

where $\langle \cdot, \cdot \rangle$ denotes cosine similarity, $\lambda_1, \lambda_2$ are weight factors, and `Embedding` is obtained via the NLP tool SpaCy. By solving the bipartite matching problem, we establish a one-to-one node matching between the predicted graph $\mathcal{G}_{\text{pred}}$ and the ground-truth graph $\mathcal{G}$.

For a predicted node $v_i$, the reward is defined as

$$
\text{Reward}(v_i) = \begin{cases}
\begin{aligned}
& \lambda_1 \cdot \langle \texttt{Embedding}(c_i), \texttt{Embedding}(\tilde{c}_j) \rangle \\
& + \lambda_2 \cdot \text{IoU}(b_i, \tilde{b}_j) \\
& + \lambda_3 \cdot \exp(-||b_i - \tilde{b}_j||_1),
\end{aligned} & \text{if } v_i \text{ and } \tilde{v}_j \text{ are matched,} \\
0, & \text{otherwise.}
\end{cases}
\tag{4}
$$

which is the linear combination of object category similarity and the IoU of bounding boxes. The total rewards of an image's prediction set $\{v_i\}_{i=1}^M$ is computed as

$$
\text{Reward}(\{v_i\}_{i=1}^M) = \frac{1}{|\mathcal{V}_{\text{gt}}|} \sum_{i=1}^M \text{Reward}(v_i).
\tag{5}
$$

For a predicted triplet $e_{ij} := < v_i, p_{ij}, v_j >$, the reward is defined as

$$
\text{Reward}(e_{ij}) = \begin{cases}
\begin{aligned}
& \langle \texttt{Embedding}(v_i), \texttt{Embedding}(\tilde{v}_k) \rangle \cdot \\
& \langle \texttt{Embedding}(v_j), \texttt{Embedding}(\tilde{v}_l) \rangle \cdot \\
& \langle \texttt{Embedding}(p_{ij}), \texttt{Embedding}(p_{kl}) \rangle,
\end{aligned} & \begin{aligned} & \text{if } v_i \text{ matches } \tilde{v}_k \\ & \text{and } v_j \text{ matches } \tilde{v}_l, \end{aligned} \\
0, & \text{otherwise.}
\end{cases}
\tag{6}
$$

Thereby, the reward of an image's predicted edge set is computed as

$$
\text{Reward}(\{e_{ij}\}) = \frac{1}{|\mathcal{E}_{\text{gt}}|} \sum \text{Reward}(e_{ij}).
\tag{7}
$$

## 4 Experiments

### 4.1 Dataset and Experiment Setup

**Dataset.** The widely-used scene graph dataset VG150 [34] consists of 150 object categories and 50 relation categories. Following prior works [40, 4], the training set used in this work contains 56,224 image-graph pairs, while the validation set includes 5,000 pairs. To prompt the M-LLM, we transform each image-graph pair using the template described in Table 6.

The Panoptic Scene Graph (PSG) dataset [36] is built on the COCO dataset [18], consisting of 80 *thing* object categories, 53 *stuff* object categories, and 56 relation categories. It contains 46,563 image-graph pairs for training and 2,186 pairs for testing.

**Evaluation.** Following the standard evaluation pipeline in SGG, we adopt the SGDET protocol [34, 30] to measure the model's ability to generate scene graphs. SGDET requires the model to generate scene graphs directly from the image without any predefined object boxes. Performance is evaluated using Recall and mean Recall (mRecall). Recall is computed for each image-graph pair, where a predicted triplet is considered correct if both the subject and object bounding boxes have an Intersection over Union (IoU) of at least 0.5 with the corresponding ground-truth boxes, and the subject category, object category, and relationship label all match the ground truth. Mean Recall (mRecall) is obtained by averaging the Recall across all relation categories. We additionally report AP@50 to assess object detection performance and Failure Rate to evaluate format consistency.

**Implementation Details.** Our code is based on the `trl` library [31] and utilizes vLLM [13] to speed up sampling during reinforcement learning. For SFT, the model is trained for 3 epochs with a batch size of 128 on 4 NVIDIA A100 (80GB) GPUs, using the AdamW optimizer [22] with a maximum learning rate of 1e-5. For RL, the model is trained for 1 epoch with a batch size of 32 and 8 generations per sample on 16 NVIDIA GH200 (120GB) GPUs, also using AdamW with a maximum learning rate of 6e-7.

### 4.2 How Well Do M-LLMs Reason About Visual Relationships?

We evaluate the visual relationship reasoning capabilities of open-source multimodal LLMs using a four-to-one Visual Question Answering (VQA) task. Each model is prompted with an image and a corresponding question. The used prompt template is: `Analyze the relationship between the object "{sub_name}" at {sub_box} and the object "{obj_name}" at {obj_box} in an image of size ({width}x{height}). The bounding boxes are in [x1, y1, x2, y2] format. Choose the most appropriate relationship from the following options: A) {choices[0]}; B) {choices[1]}; C) {choices[2]}; D) {choices[3]}.` We report Acc (accuracy over all questions) and mAcc (mean accuracy per image) in Table 7. The results reveal that many multimodal LLMs struggle with visual relationship reasoning. Moreover, the task exhibits a noticeable text bias, and the presence of bounding boxes can sometimes mislead the model's attention. As a simpler task compared to SGG, the poor performance suggests that directly applying multimodal LLMs to SGG may yield suboptimal results.

### 4.3 How Well do M-LLMs Generate Scene Graphs?

#### 4.3.1 Benchmark on VG150

We report the performance under various settings in Table 1, which includes: 1) *Specific Models*: Methods built on specific detectors such as Faster R-CNN [26] (e.g., IMP [34]) or DETR [1] (e.g., OvSGTR [4]) for scene graph generation. 2) *Commercial M-LLMs*: Advanced multimodal large language models such as GPT-4o [9] and Gemini 1.5 Flash [25]. 3) *Open-source M-LLMs*: Publicly available models such as LLaVA v1.5 [21], Qwen2-VL [32], and our proposed *R1-SGG-Zero* (based on `Qwen2-VL-2B/7B-Instruct`, trained with GRPO but without supervised fine-tuning) and *R1-SGG* (built on the same backbone, fine-tuned with GRPO and initialized from SFT checkpoints).

The results in Table 1 reveal several key observations.

**Zero-shot Performance of M-LLMs.** Either commercial or open-source multimodal LLMs struggle to generate accurate scene graphs, and this can be attributed to several factors. First, the internal processing of private models such as GPT-4o remains opaque to users, resulting in suboptimal

Table 1: SGDET performance on the VG150 validation set. For M-LLMs, predefined object classes and relation categories are included in the input prompts.

| Method | Params | Failure Rate (%) | AP@50 | Recall | mRecall |
|---|---|---|---|---|---|
| *Specific Models* | | | | | |
| IMP [34] | | | 20.91 | 17.85 | 2.66 |
| MOTIFS [38] | - | - | 29.56 | 27.21 | 7.84 |
| VCTree [29] | | | 28.13 | 24.87 | 8.47 |
| OvSGTR [4] | | | 33.39 | 26.74 | 5.83 |
| *Commercial M-LLMs* | | | | | |
| GPT-4o [9] | - | 2.94 | 0.00 | 0.00 | 0.00 |
| Gemini 1.5 Flash [25] | - | 1.10 | 0.51 | 0.10 | 0.08 |
| Gemini 2.0 Flash [6] | - | 1.06 | 0.54 | 0.07 | 0.03 |
| *Open-sourced M-LLMs* | | | | | |
| LLaVA v1.5 [21] | 7B | 82.70 | 0.00 | 0.00 | 0.00 |
| Qwen2-VL-2B-Instruct [32] | 2B | 59.96 | 2.18 | 0.07 | 0.18 |
| +SFT | 2B | 72.42 | 8.10 | 5.47 | 1.46 |
| Qwen2-VL-7B-Instruct [32] | 7B | 54.46 | 6.07 | 0.69 | 0.80 |
| +SFT | 7B | 39.54 | 14.18 | 9.62 | 3.30 |
| R1-SGG-Zero | 2B | 0.34 | 12.30 | 11.89 | 5.70 |
| R1-SGG | 2B | 0.10 | 17.87 | 21.09 | 7.48 |
| R1-SGG-Zero | 7B | 0.04 | 15.59 | 18.34 | 8.32 |
| R1-SGG | 7B | **0.08** | **19.47** | **23.75** | **11.43** |

object detection performance. Second, models like LLaVA v1.5 align visual and textual features only at the image level, typically using a fixed resolution of $336 \times 336$, which restricts spatial understanding. Third, although models such as Gemini 2.0 and Qwen2-VL demonstrate a degree of spatial understanding, the task of scene graph generation is much complex than pure object detection or visual grounding. Consequently, their zero-shot performance drops significantly.

**SFT vs. RL. 1)** RL substantially improves performance across all metrics compared to SFT alone. Specifically, RL dramatically reduces the failure rate (*e.g.*, from 72.42% to 0.10% for 2B models) and yields significant gains in AP@50, Recall, and mRecall. This highlights the effectiveness of GRPO in enhancing the model's ability to generate accurate and complete scene graphs. **2)** SFT achieves moderate improvements in AP@50 and Recall over the baseline but struggles with a relatively high failure rate. This suggests that SFT primarily improves relation prediction while being less effective at correcting structural errors, such as missing objects, relationships, or format inconsistencies. **3)** applying RL on top of SFT (*i.e.*, R1-SGG) further boosts performance over both SFT and R1-SGG-Zero in most cases. This indicates that combining SFT and RL benefits from better initialization, leading to stronger relation recognition and higher recall. **4)** larger models (*e.g.*, 7B) consistently outperform smaller models (*e.g.*, 2B) across AP@50, Recall, and mRecall, demonstrating the benefits of scaling model capacity for scene graph generation.

**Compared to Specific Models.** The gap between AP@50 and Recall highlights the advantage of dense predictions. However, our models, such as *R1-SGG*, achieve a notable mean Recall (mRecall) of 11.43%, suggesting that multimodal LLMs are more effective at generating less biased scene graphs. Moreover, specific models are typically restricted to a limited vocabulary and struggle to generalize across domains, whereas multimodal LLMs exhibit greater adaptability and broader generalization capabilities.

Overall, the results demonstrate that reinforcement learning (RL) significantly reduces the failure rate and enhances both object detection and relationship recognition. In contrast, supervised fine-tuning (SFT) alone results in a relatively high failure rate and limited improvements. As shown in Fig. 2, the failure rate quickly drops to near-zero with RL, whereas SFT continues to suffer from frequent structural errors.

### 4.3.2 Benchmark on PSG

As shown in Table 2, our R1-SGG approach achieves strong performance on the PSG dataset. Compared to baselines, SFT significantly improves AP@50, Recall, and mean Recall (mRecall), while reinforcement learning further enhances relationship recognition, achieving the highest Recall (43.48% for 7B model) and mRecall (33.71%). Notably, our method also drives the failure rate to

Table 2: Performance on the PSG dataset [36]. For M-LLMs, predefined object classes and relation categories are included in the input prompts.

| Model | Params | Failure Rate (%) | AP@50 | Recall | mRecall |
|---|---|---|---|---|---|
| *Specific Models* | | | | | |
| IMP [34] | | | | 16.50 | 6.50 |
| MOTIFS [38] | | | | 20.00 | 9.10 |
| VCTree [29] | - | - | - | 20.60 | 9.70 |
| GPSNet [19] | | | | 17.80 | 7.00 |
| PSGFormer [36] | | | | 18.60 | 16.70 |
| *Open-sourced M-LLMs* | | | | | |
| LLaVA v1.5 [21] | 7B | 81.97 | 0.07 | 0.00 | 0.00 |
| TextPSG [42] | - | - | - | 4.80 | - |
| ASMv2 [33] | 13B | 0.87 | 21.45 | 14.77 | 11.82 |
| LLaVA-SpaceSGG [35] | 13B | - | - | 15.43 | 13.23 |
| Qwen2-VL-2B-Instruct | 2B | 67.20 | 4.89 | 0.39 | 0.26 |
| +SFT | 2B | 6.54 | 36.05 | 22.06 | 14.92 |
| Qwen2-VL-7B-Instruct | 7B | 37.97 | 12.75 | 3.18 | 4.33 |
| +SFT | 7B | 0.96 | 40.79 | 24.73 | 17.11 |
| R1-SGG-Zero | 2B | 0.23 | 25.61 | 25.06 | 18.15 |
| R1-SGG | 2B | 2.70 | 39.28 | 38.49 | 31.21 |
| R1-SGG-Zero | 7B | 0.00 | 32.92 | 37.00 | 32.04 |
| R1-SGG | 7B | **0.00** | **42.05** | **43.48** | **33.71** |

zero, demonstrating the effectiveness of reinforcement learning in promoting structured, accurate scene graph generation even without predefined object categories.

## 4.4 Qualitative Results

We present qualitative results in Fig. 6 and Fig. 7. As shown in Fig. 6, the ground-truth scene graph (Fig. 6-(a)) captures key objects and their relationships but is biased toward the predicate "*has*". Conversely, the zero-shot Qwen2-VL-7B-Instruct (Fig. 6-(b)) fails to generate a valid JSON output, indicating poor instruction-following ability. With supervised fine-tuning, the model produces structurally valid graphs (Fig. 6-(c)) but frequently omits important relationships, resulting in a sparse scene graph. R1-SGG-Zero (7B), trained with RL only, improves relational recall and structure (Fig. 6-(d)), yet still outputs inaccurate triplets such as *<wheel, on, horse>* and *<helmet.2, on, horse>*. Finally, R1-SGG (7B), trained with both SFT and RL, produces a complete and consistent scene graph (Fig. 6-(e)), with results that even surpass the ground truth in relational richness.

## 4.5 Discussion

Through the exploration of applying GRPO to the SGG task, we make several observations.

**KL Regularization.** We compare models trained with and without KL divergence regularization in Fig. 5. From the result, removing KL regularization leads to improved performance, particularly with a significant reduction in failure rate.

**Sampling Length.** In our experiments, the default sampling length is set to 1,024, which sufficiently covers most corrected answers. As shown in Fig. 5, increasing the sampling length to 2,048 does not yield further performance improvements, suggesting that longer sampling might enlarge the search space and introduce additional optimization difficulties without clear benefits. This observation aligns with prior findings on test-time scaling, where increasing Chain-of-Thought (CoT) length can degrade performance [39].

**Group Size.** As shown in Fig. 5, increasing the group size from 8 to 16 stabilizes training performance, consistent with the intuition that more candidates reduce variance in group statistics estimation. To balance computational cost and performance, we adopt a group size of 8 as the default in this work.

**To Think or Not Think?** We adopt the `<think>⋯</think><answer>⋯</answer>` format in the system prompt, following DeepSeek R1 [8]. However, models such as `Qwen2-VL-2B/7B-Instruct` often fail to produce outputs with the `<think>` tag after fine-tuning, indicating difficulty in adhering to the intended structure. This suggests that rule-based rewards alone

Table 3: Generalization across datasets using `Qwen2-VL-7B-Instruct` as the baseline. Columns under *Pre-training* indicate whether the weights were initialized from specific checkpoints, while the *Training* column specifies the dataset(s) used during the fine-tuning stage. "*w/o* cats." denotes prompts without predefined object classes or relation categories.

| Model | Pre-Training | Training | VG150 | | | | PSG | | | |
|---|---|---|---|---|---|---|---|---|---|---|
| | | | Failure Rate | AP@50 | Recall | mRecall | Failure Rate | AP@50 | Recall | mRecall |
| baseline | - | - | 54.46 | 6.07 | 0.69 | 0.80 | 37.97 | 12.75 | 3.18 | 4.33 |
| baseline (*w/o* cats.) | - | - | 44.58 | 6.83 | 0.61 | 0.37 | 30.28 | 13.79 | 1.96 | 2.30 |
| SFT | - | VG150 | 39.54 | 14.18 | 9.62 | 3.30 | 22.10 | 11.05 | 3.03 | 1.36 |
| SFT (*w/o* cats.) | - | VG150 | 42.98 | 13.03 | 8.94 | 2.47 | 19.81 | 12.15 | 3.87 | 1.81 |
| R1-SGG-Zero | - | VG150 | 0.04 | 15.59 | 18.34 | 8.32 | **0.18** | **24.92** | **13.83** | **8.90** |
| R1-SGG-Zero (*w/o* cats.) | - | VG150 | 0.06 | 15.30 | 16.33 | 6.94 | 0.18 | 18.10 | 6.16 | 3.38 |
| R1-SGG | SFT | VG150 | **0.08** | **19.47** | **23.75** | **11.43** | 0.23 | 18.12 | 9.10 | 5.13 |
| R1-SGG (*w/o* cats.) | SFT (*w/o* cats.) | VG150 | 0.30 | 18.09 | 22.73 | 9.62 | 0.64 | 14.64 | 7.51 | 3.88 |
| SFT | - | PSG | 36.98 | 5.79 | 1.42 | 0.77 | 0.91 | 40.58 | 24.75 | 17.31 |
| SFT (*w/o* cats.) | - | PSG | 2.54 | 7.94 | 1.77 | 1.25 | 1.01 | 39.02 | 23.70 | 17.17 |
| R1-SGG-Zero | - | PSG | **0.12** | **14.22** | **8.90** | **5.34** | 0.00 | 32.92 | 37.00 | 32.04 |
| R1-SGG-Zero (*w/o* cats.) | - | PSG | 0.02 | 9.08 | 2.80 | 1.78 | 0.05 | 24.26 | 19.94 | 18.04 |
| R1-SGG | SFT | PSG | 0.94 | 10.38 | 4.40 | 2.69 | **0.00** | **42.05** | **43.48** | **33.71** |
| R1-SGG (*w/o* cats.) | SFT (*w/o* cats.) | PSG | 0.14 | 9.38 | 2.16 | 1.55 | 0.14 | 41.15 | 41.44 | 31.51 |

Table 4: Ablation of reward formulations on VG150 validation set using R1-SGG (7B).

| Setting | Sparsity | Metric Aligned | Failure Rate (%) | AP@50 | Recall (%) | mRecall (%) |
|---|---|---|---|---|---|---|
| Hard Recall | sparse | ✓ | 0.08 | 19.47 | 23.75 | 11.43 |
| Hard Recall + Relax | medium | ✗ | 0.02 | 19.93 | 24.05 | 9.61 |
| Soft Recall | dense | ✗ | 0.06 | 18.73 | 21.92 | 5.61 |

are insufficient to trigger abstract reasoning patterns like CoT, and highlights the need for additional SFT on CoT-specific datasets to incentivize coherent intermediate reasoning.

**Generalization Across Datasets.** We report performance comparisons across datasets in Table 3. The results highlight several key insights: 1) **VG150 poses a significantly greater challenge than PSG.** For instance, SFT trained solely on PSG achieves a high AP@50 of 40.58 and Recall of 24.75%, with a low failure rate of 0.91%. In contrast, SFT trained only on VG150 results in a much higher failure rate of 39.54%, with notably lower AP@50 (14.18) and Recall (9.62%). 2) **SFT has a strong domain-specific effect.** SFT models trained on one dataset (*e.g.*, VG150) exhibit substantial performance drops when evaluated on another (*e.g.*, PSG), reflecting limited transferability. For example, VG150-trained SFT only achieves 3.03% Recall and 1.36% mRecall on PSG. 3) **Predefined categories in the prompt.** Models trained and evaluated without categories (denoted as "*w/o* cats.") generally exhibit a slight drop in performance, while those with category information demonstrate better generalization under open-set settings. 4) **Initialization of RL matters.** R1-SGG initialized with SFT checkpoints consistently outperforms R1-SGG-Zero. On VG150, R1-SGG (7B) achieves 23.75% Recall and 11.43% mRecall versus 18.34% and 8.32% for R1-SGG-Zero. A similar trend is observed on PSG. This highlights the importance of using SFT as a warm-start for reinforcement learning, which leads to improved sample efficiency and stronger downstream performance. 5) **R1-SGG-Zero exhibits stronger cross-dataset generalization.** This aligns with the domain-specific nature of SFT—models trained via SFT tend to overfit to the source domain, resulting in degraded performance on unseen datasets. In contrast, R1-SGG-Zero, trained without SFT, generalizes more robustly across domains.

**Hard Recall vs. Soft Recall.** As shown in Table 4, *Hard Recall* outperforms other variants despite providing sparser reward signals. This highlights the importance of aligning reward functions with evaluation metrics, rather than prioritizing reward smoothness alone.

# 5 Conclusion

We present a reinforcement learning framework for enhancing end-to-end Scene Graph Generation (SGG) with multimodal large language models (M-LLMs). To align training with the structured nature of scene graphs, we design a set of rule-based rewards, comprising three scene graph variants (*Hard Recall*, *Hard Recall+Relax*, and *Soft Recall*) and a format consistency reward, which enable fine-grained and stable policy optimization via GRPO. Our approach significantly improves the structural validity and relational accuracy of generated scene graphs. We release our code and models to support future research on structured visual understanding with M-LLMs.

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

Table 5: Prompting an M-LLM to generate scene graphs without providing predefined object classes or predicate types.

> messages = [{ **"role"**: "system", **"content"**: " {system_prompt}" }, { **"role"**: "user",
>
> **"content"**: f"""Generate a structured scene graph for an image using the following format: "'json { "objects": [ {"id": "object_name.number", "bbox": [x1, y1, x2, y2]}, ... ], "relationships": [ {"subject": "object_name.number", "predicate": "relationship_type", "object": "object_name.number"}, ... ] }"'. ### **Guidelines:** - **Objects:** - Assign a unique ID for each object using the format "object_name.number" (e.g., "person.1", "bike.2"). - Provide its bounding box '[x1, y1, x2, y2]' in integer pixel format. - Include all visible objects, even if they have no relationships.
> - **Relationships:** - Represent interactions accurately using "subject", "predicate", and "object". - Omit relationships for orphan objects.
> ### **Example Output:** "'json { "objects": [ {"id": "person.1", "bbox": [120, 200, 350, 700]}, {"id": "bike.2", "bbox": [100, 600, 400, 800]}, {"id": "helmet.3", "bbox": [150, 150, 280, 240]}, {"id": "tree.4", "bbox": [500, 100, 750, 700]} ], "relationships": [ {"subject": "person.1", "predicate": "riding", "object": "bike.2"}, {"subject": "person.1", "predicate": "wearing", "object": "helmet.3"} ] } "' Now, generate the complete scene graph for the provided image: """ } ]

Table 6: Prompting an M-LLM to generate scene graphs with predefined object classes and predicate types. Here, *OBJ_CLS* and *REL_CLS* refer to the predefined object classes and relation categories respectively.

> messages = [{ **"role"**: "system", **"content"**: " {system_prompt}" }, { **"role"**: "user",
>
> **"content"**: f"""Generate a structured scene graph for an image using the following format: "'json { "objects": [ {"id": "object_name.number", "bbox": [x1, y1, x2, y2]}, ... ], "relationships": [ {"subject": "object_name.number", "predicate": "relationship_type", "object": "object_name.number"}, ... ] }"'. ### **Guidelines:** - **Objects:** - Assign a unique ID for each object using the format "object_name.number" (e.g., "person.1", "bike.2"). The **object_name** must belong to the predefined object set: '{OBJ_CLS}'. - Provide its bounding box '[x1, y1, x2, y2]' in integer pixel format. - Include all visible objects, even if they have no relationships.
> - **Relationships:** - Represent interactions accurately using "subject", "predicate", and "object". - Omit relationships for orphan objects. The **predicate** must belong to the predefined relationship set: '{REL_CLS}'. ### **Example Output:** "'json { "objects": [ {"id": "person.1", "bbox": [120, 200, 350, 700]}, {"id": "bike.2", "bbox": [100, 600, 400, 800]}, {"id": "helmet.3", "bbox": [150, 150, 280, 240]}, {"id": "tree.4", "bbox": [500, 100, 750, 700]} ], "relationships": [ {"subject": "person.1", "predicate": "riding", "object": "bike.2"}, {"subject": "person.1", "predicate": "wearing", "object": "helmet.3"} ] } "' Now, generate the complete scene graph for the provided image: """ } ]

# A    Supplementary Material

## A.1    Prompt Templates for SGG

In this work, we adopt two prompt templates for scene graph generation, as illustrated in Table 6 and Table 5. The difference lies in whether predefined object classes and relation categories are provided.

## A.2    How Well Do M-LLMs Reason About Visual Relationships?

To evaluate the reasoning capabilities of M-LLMs over visual relationships, we present results in Table 7. We vary both the visual input and the text prompt conditions to assess robustness. For visual variations, we consider: *org. img.*, *mask img.*, and *mask obj.*; for prompt variations, we add: *w/o cats.* (without object categories) and *w/o box.* (without bounding boxes).

Table 7: Comparison of VQA on the VG150 validation set across various models and settings. Gains compared to the *Original Image* (1st row) are indicated in red. *"mask img."* refers to masking the entire image with random noise, *"mask obj."* refers to masking object regions with black pixels, *"w/o cats."* refers to not providing object categories in the prompt, and *"w/o box."* refers to not providing bounding boxes in the prompt.

| | InstructBLIP 7B | | LLaVA v1.5 7B | | LLaVA v1.6 7B | | Qwen2VL 7B | |
| --- | --- | --- | --- | --- | --- | --- | --- | --- |
| | Acc | mAcc | Acc | mAcc | Acc | mAcc | Acc | mAcc |
| org. img. | 2.3 | 1.9 | 45.8 | 45.6 | 28.7 | 29.2 | 53.7 | 53.4 |
| mask img. | 1.0 (-1.3) | 1.0 (-0.9) | 21.8 (-24.0) | 21.6 (-24.0) | 3.9 (-24.8) | 4.0 (-25.2) | 0.0 (-53.7) | 0.0 (-53.4) |
| mask obj. | 1.9 (-0.4) | 1.9 (-0.1) | 37.2 (-8.6) | 37.2 (-8.4) | 12.8 (-15.9) | 13.2 (-16.0) | 16.2 (-37.5) | 16.8 (-36.5) |
| w/o cats. | 2.5 (+0.2) | 2.4 (+0.4) | 32.8 (-12.9) | 32.7 (-12.9) | 9.5 (-19.2) | 10.1 (-19.1) | 16.8 (-36.9) | 18.1 (-35.3) |
| + mask img. | 1.0 (-1.3) | 1.0 (-0.9) | 15.4 (-30.3) | 15.3 (-30.3) | 0.0 (-28.7) | 0.0 (-29.2) | 0.2 (-53.6) | 0.2 (-53.1) |
| + mask obj. | 1.8 (-0.5) | 1.7 (-0.3) | 27.9 (-17.8) | 28.4 (-17.2) | 3.3 (-25.4) | 3.8 (-25.4) | 4.7 (-49.1) | 5.5 (-47.8) |
| w/o box. | 26.0 (+23.7) | 25.9 (+24.0) | 61.9 (+16.2) | 61.3 (+15.7) | 53.5 (+24.8) | 52.1 (+22.9) | 78.1 (+24.4) | 77.1 (+23.8) |
| + mask img. | 10.1 (+7.9) | 10.2 (+8.2) | 36.3 (-9.5) | 35.2 (-10.4) | 11.5 (-17.2) | 11.4 (-17.7) | 0.0 (-53.7) | 0.0 (-53.4) |
| + mask obj. | 19.3 (+17.0) | 19.1 (+17.1) | 54.2 (+8.5) | 53.8 (+8.2) | 33.5 (+4.8) | 33.2 (+4.1) | 40.3 (-13.4) | 39.3 (-14.1) |

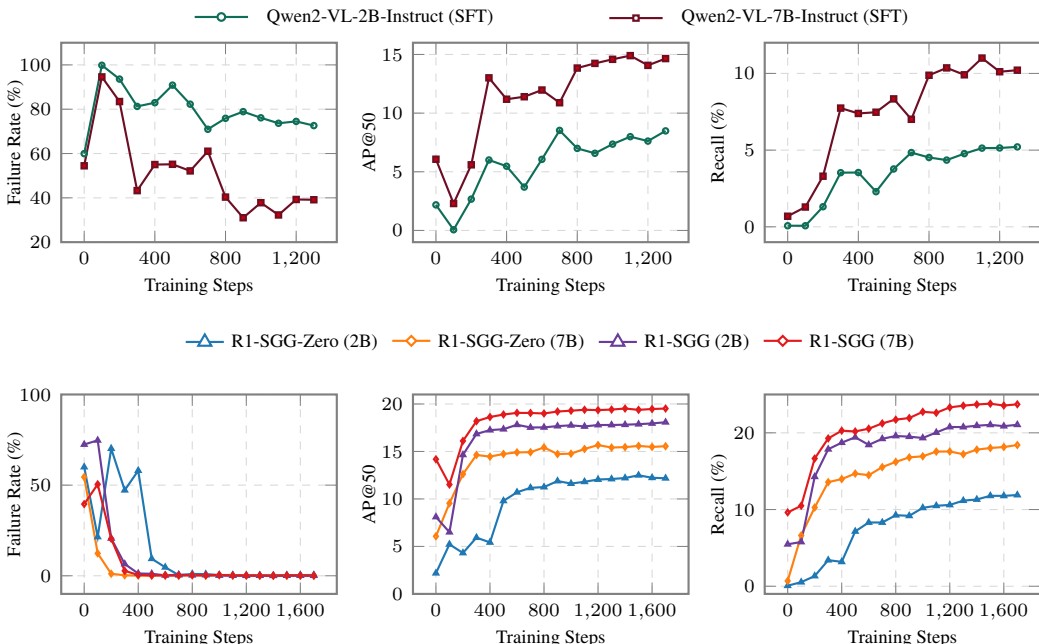

Figure 2: Comparison of R1-SGG-Zero and R1-SGG models against SFT baselines (Qwen2-VL-2B/7B-Instruct) across training steps on the VG150 validation set in terms of Failure Rate (%), AP@50, and Recall (%).

## A.3 Qualitative Results

We present qualitative results in Fig. 6 and Fig. 7, and analyze head and tail predicate performance in Fig. 3 and Fig. 4 to assess long-tail bias. As shown in Fig. 3, both specific models such as OvSGTR and M-LLMs like Qwen2-VL-7B-Instruct (with or without SFT) tend to be biased toward head classes, whereas R1-SGG achieves significantly higher recall on tail predicates. This trend is also confirmed on the PSG dataset in Fig. 4. These results demonstrate that R1-SGG is more effective at generating unbiased scene graphs.

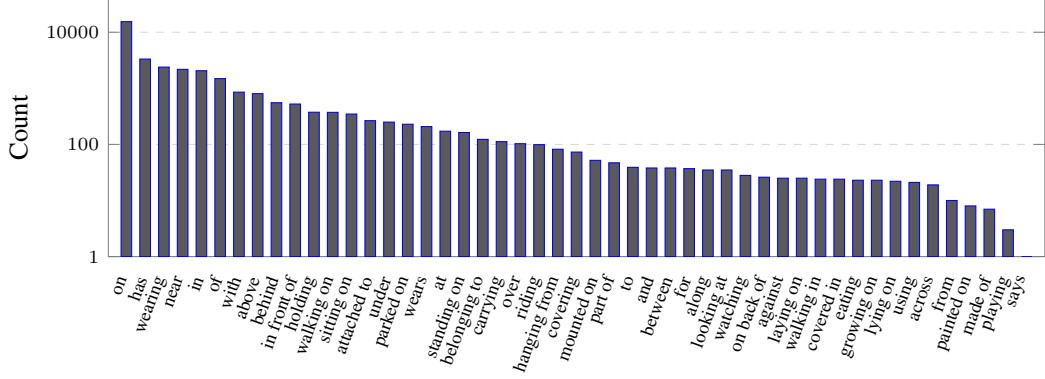

(a) Histogram of predicate frequency in the VG150 validation set.

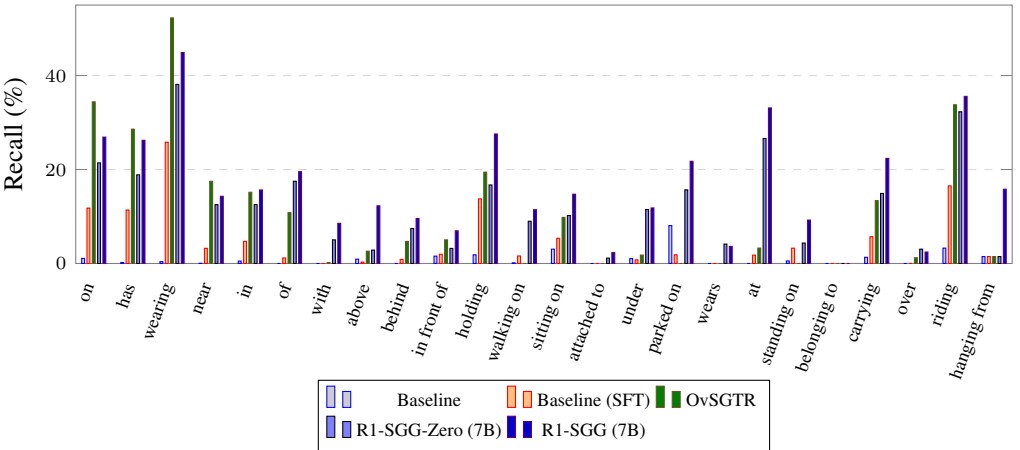

(b) Recall scores of top-24 predicates of the VG150 validation set.

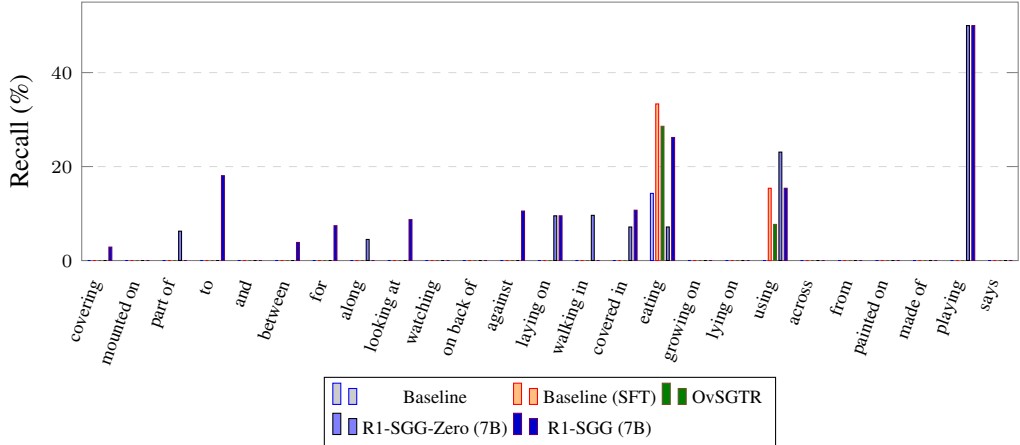

(c) Recall scores of tail-25 predicates of the VG150 validation set.

Figure 3: Comparison of predicate frequency and predicate-wise recall on the VG150 validation set. Subfigures (b) and (c) report the recall performance of *R1-SGG* compared to four models on the top-24 and tail-25 predicates (the VG150 validation set contains only 49 predicates, with the predicate "*flying in*" missing.), respectively. Here, *Baseline* refers to Qwen2-VL-7B-Instruct.

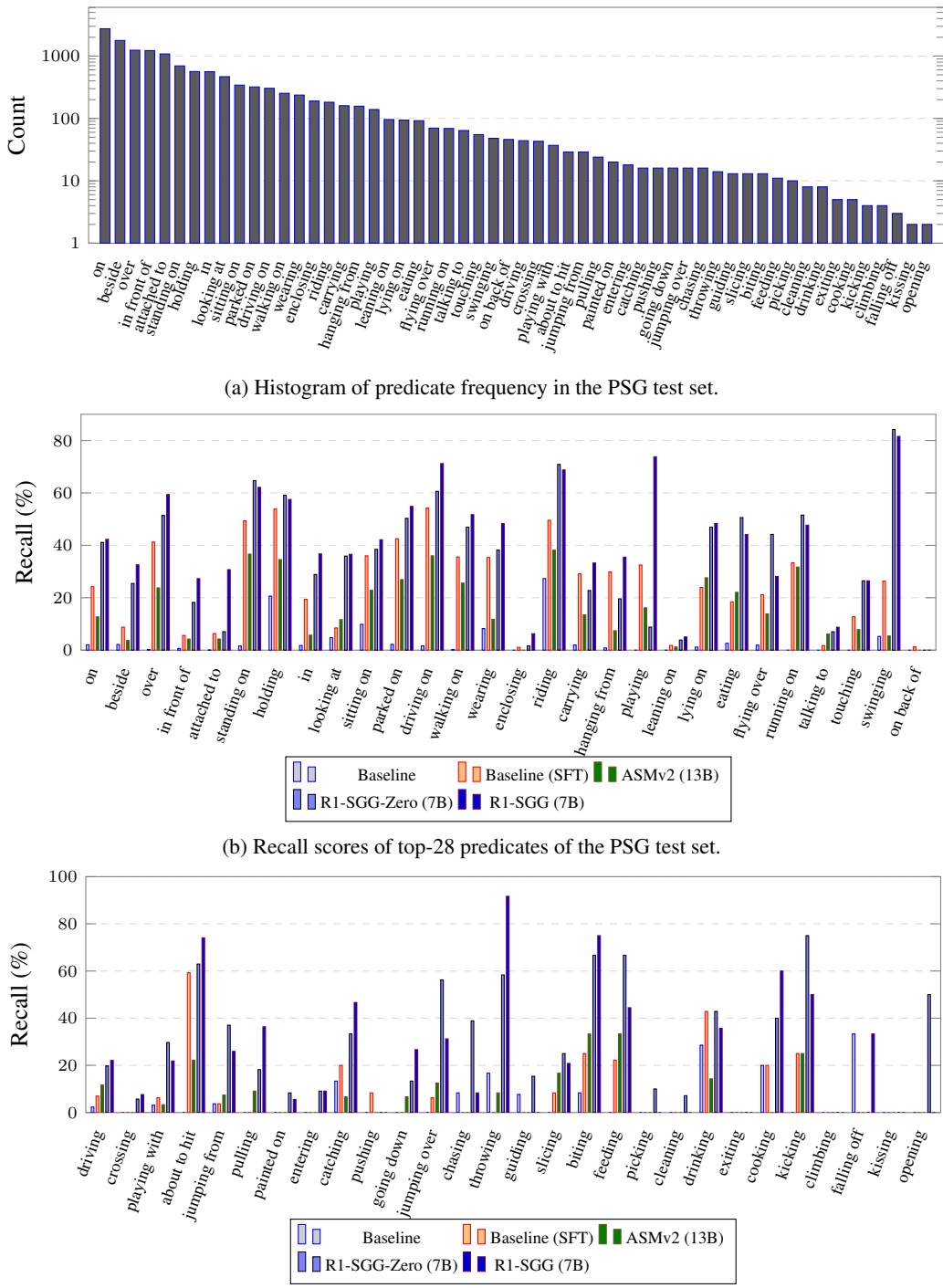

(a) Histogram of predicate frequency in the PSG test set.

(b) Recall scores of top-28 predicates of the PSG test set.

(c) Recall scores of tail-28 predicates of the PSG test set.

Figure 4: Comparison of predicate frequency and predicate-wise recall on the PSG test set. Subfigures (b) and (c) report the recall performance of *R1-SGG* compared to four models on the top-28 and tail-28 predicates, respectively. Here, *Baseline* refers to `Qwen2-VL-7B-Instruct`.

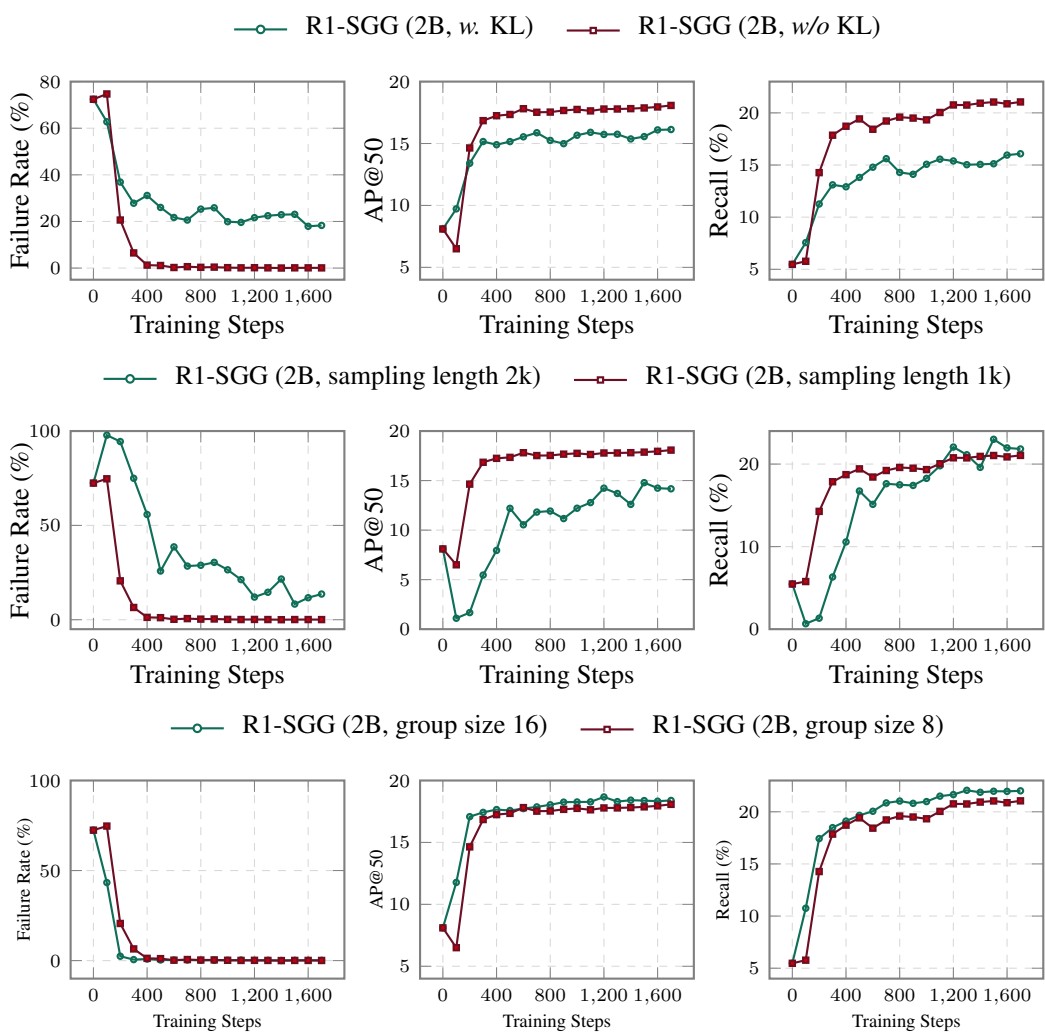

Figure 5: Performance comparison of R1-SGG (2B) across training steps on the VG150 validation set. Each row evaluates a different setting: (Top) KL divergence regularization ($\beta$=0.04 vs. $\beta$=0), (Middle) sampling length, and (Bottom) group size. Metrics reported include Failure Rate (%), AP@50, and Recall (%).


Figure 6: Qualitative comparison of generated scene graphs (from VG150). (a) Ground-truth scene graph annotated by humans. (b) Zero-shot Qwen2-VL-7B-Instruct produces an invalid JSON (failure to follow format). (c) Qwen2-VL-7B-Instruct (SFT) outputs a valid graph but omits many relations. (d) R1-SGG-Zero (7B) recovers most objects and relations but still hallucinates incorrect triplets (*e.g.*, < *wheel, on, horse*> and < *helmet.2, on, horse* >). (e) R1-SGG (7B) yields a complete, structurally correct scene graph with higher recall.

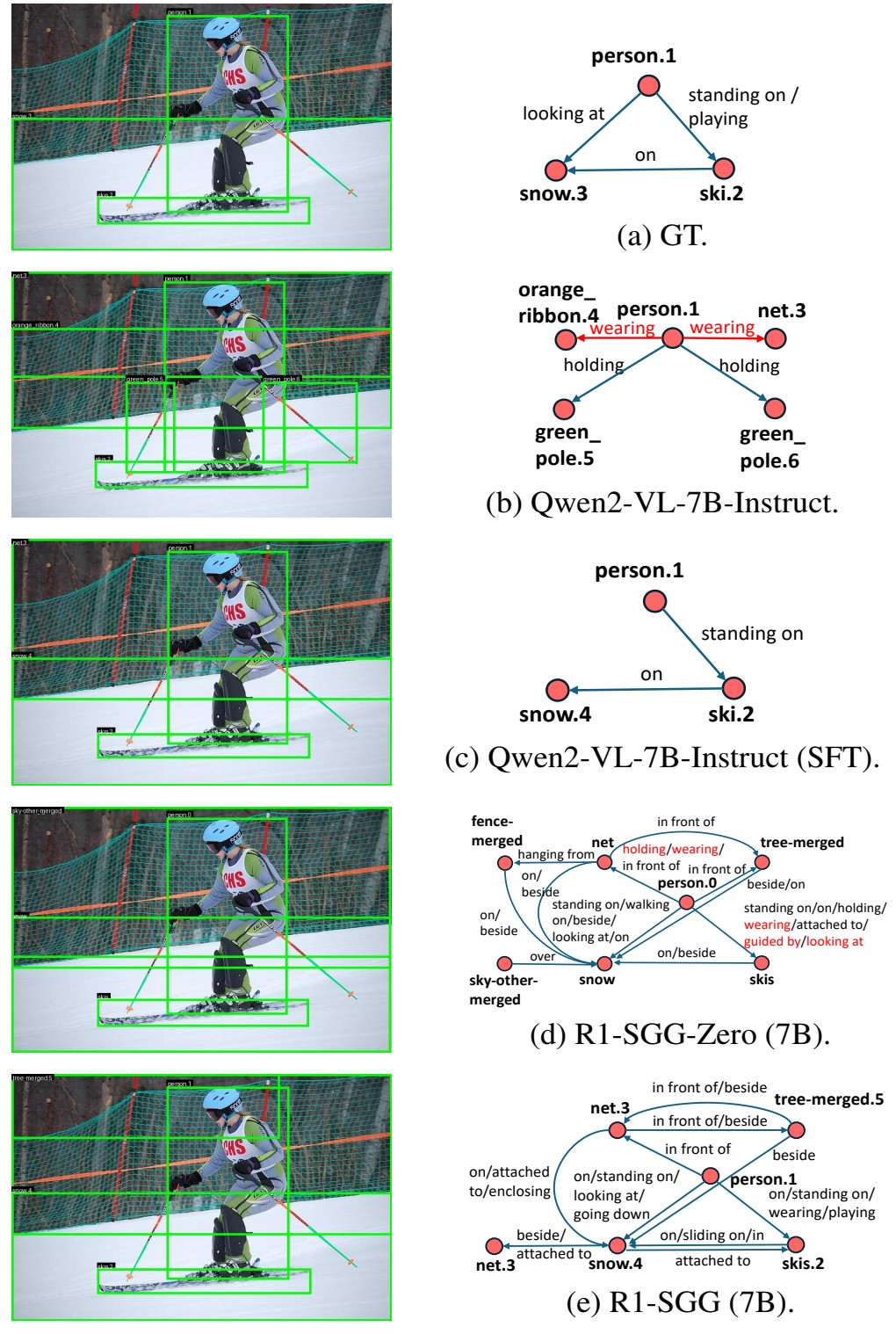

Figure 7: Qualitative comparison of generated scene graphs (from PSG). (a) Ground-truth scene graph annotated by humans. (b) Zero-shot Qwen2-VL-7B-Instruct generates a valid graph but includes incorrect triplets (*e.g.*, *<person.1, wearing, net.3>*). (c) Qwen2-VL-7B-Instruct (SFT) produces a valid graph but omits some relationships. (d) R1-SGG-Zero (7B) recovers most objects and relations but still hallucinates errors (*e.g.*, *<person.0, wearing, net>*). (e) R1-SGG (7B) generates a complete and accurate scene graph with higher recall.

- The abstract and/or introduction should clearly state the claims made, including the contributions made in the paper and important assumptions and limitations. A No or NA answer to this question will not be perceived well by the reviewers.
- The claims made should match theoretical and experimental results, and reflect how much the results can be expected to generalize to other settings.
- It is fine to include aspirational goals as motivation as long as it is clear that these goals are not attained by the paper.

2. **Limitations**

Question: Does the paper discuss the limitations of the work performed by the authors?

Answer: [Yes]

Justification: Limitations are discussed, particularly in Section 4.5.

Guidelines:

- The answer NA means that the paper has no limitation while the answer No means that the paper has limitations, but those are not discussed in the paper.
- The authors are encouraged to create a separate "Limitations" section in their paper.
- The paper should point out any strong assumptions and how robust the results are to violations of these assumptions (e.g., independence assumptions, noiseless settings, model well-specification, asymptotic approximations only holding locally). The authors should reflect on how these assumptions might be violated in practice and what the implications would be.
- The authors should reflect on the scope of the claims made, e.g., if the approach was only tested on a few datasets or with a few runs. In general, empirical results often depend on implicit assumptions, which should be articulated.
- The authors should reflect on the factors that influence the performance of the approach. For example, a facial recognition algorithm may perform poorly when image resolution is low or images are taken in low lighting. Or a speech-to-text system might not be used reliably to provide closed captions for online lectures because it fails to handle technical jargon.
- The authors should discuss the computational efficiency of the proposed algorithms and how they scale with dataset size.
- If applicable, the authors should discuss possible limitations of their approach to address problems of privacy and fairness.
- While the authors might fear that complete honesty about limitations might be used by reviewers as grounds for rejection, a worse outcome might be that reviewers discover limitations that aren't acknowledged in the paper. The authors should use their best judgment and recognize that individual actions in favor of transparency play an important role in developing norms that preserve the integrity of the community. Reviewers will be specifically instructed to not penalize honesty concerning limitations.

3. **Theory assumptions and proofs**

Question: For each theoretical result, does the paper provide the full set of assumptions and a complete (and correct) proof?

Answer: [NA]

Justification: This is not a theoretical paper.

Guidelines:

- The answer NA means that the paper does not include theoretical results.
- All the theorems, formulas, and proofs in the paper should be numbered and cross-referenced.
- All assumptions should be clearly stated or referenced in the statement of any theorems.
- The proofs can either appear in the main paper or the supplemental material, but if they appear in the supplemental material, the authors are encouraged to provide a short proof sketch to provide intuition.
- Inversely, any informal proof provided in the core of the paper should be complemented by formal proofs provided in appendix or supplemental material.

- Theorems and Lemmas that the proof relies upon should be properly referenced.

4. **Experimental result reproducibility**

   Question: Does the paper fully disclose all the information needed to reproduce the main experimental results of the paper to the extent that it affects the main claims and/or conclusions of the paper (regardless of whether the code and data are provided or not)?

   Answer: [Yes]

   Justification: The paper describes datasets, training setup, reward design, hyperparameters, model configurations, and evaluation metrics. This level of detail supports reproducibility even without direct code access.

   Guidelines:

   - The answer NA means that the paper does not include experiments.
   - If the paper includes experiments, a No answer to this question will not be perceived well by the reviewers: Making the paper reproducible is important, regardless of whether the code and data are provided or not.
   - If the contribution is a dataset and/or model, the authors should describe the steps taken to make their results reproducible or verifiable.
   - Depending on the contribution, reproducibility can be accomplished in various ways. For example, if the contribution is a novel architecture, describing the architecture fully might suffice, or if the contribution is a specific model and empirical evaluation, it may be necessary to either make it possible for others to replicate the model with the same dataset, or provide access to the model. In general. releasing code and data is often one good way to accomplish this, but reproducibility can also be provided via detailed instructions for how to replicate the results, access to a hosted model (e.g., in the case of a large language model), releasing of a model checkpoint, or other means that are appropriate to the research performed.
   - While NeurIPS does not require releasing code, the conference does require all submissions to provide some reasonable avenue for reproducibility, which may depend on the nature of the contribution. For example
     (a) If the contribution is primarily a new algorithm, the paper should make it clear how to reproduce that algorithm.
     (b) If the contribution is primarily a new model architecture, the paper should describe the architecture clearly and fully.
     (c) If the contribution is a new model (e.g., a large language model), then there should either be a way to access this model for reproducing the results or a way to reproduce the model (e.g., with an open-source dataset or instructions for how to construct the dataset).
     (d) We recognize that reproducibility may be tricky in some cases, in which case authors are welcome to describe the particular way they provide for reproducibility. In the case of closed-source models, it may be that access to the model is limited in some way (e.g., to registered users), but it should be possible for other researchers to have some path to reproducing or verifying the results.

5. **Open access to data and code**

   Question: Does the paper provide open access to the data and code, with sufficient instructions to faithfully reproduce the main experimental results, as described in supplemental material?

   Answer: [Yes]

   Justification: We use public datasets and the code will be released. Supplementary material also provides detailed prompt templates for reproducibility.

   Guidelines:

   - The answer NA means that paper does not include experiments requiring code.
   - Please see the NeurIPS code and data submission guidelines (`https://nips.cc/public/guides/CodeSubmissionPolicy`) for more details.

- While we encourage the release of code and data, we understand that this might not be possible, so "No" is an acceptable answer. Papers cannot be rejected simply for not including code, unless this is central to the contribution (e.g., for a new open-source benchmark).
- The instructions should contain the exact command and environment needed to run to reproduce the results. See the NeurIPS code and data submission guidelines (`https://nips.cc/public/guides/CodeSubmissionPolicy`) for more details.
- The authors should provide instructions on data access and preparation, including how to access the raw data, preprocessed data, intermediate data, and generated data, etc.
- The authors should provide scripts to reproduce all experimental results for the new proposed method and baselines. If only a subset of experiments are reproducible, they should state which ones are omitted from the script and why.
- At submission time, to preserve anonymity, the authors should release anonymized versions (if applicable).
- Providing as much information as possible in supplemental material (appended to the paper) is recommended, but including URLs to data and code is permitted.

6. **Experimental setting/details**

Question: Does the paper specify all the training and test details (e.g., data splits, hyper-parameters, how they were chosen, type of optimizer, etc.) necessary to understand the results?

Answer: [Yes]

Justification: Section 4.1 clearly provides training details, dataset splits, evaluation protocols, model sizes, batch sizes, learning rates, and compute resources used.

Guidelines:

- The answer NA means that the paper does not include experiments.
- The experimental setting should be presented in the core of the paper to a level of detail that is necessary to appreciate the results and make sense of them.
- The full details can be provided either with the code, in appendix, or as supplemental material.

7. **Experiment statistical significance**

Question: Does the paper report error bars suitably and correctly defined or other appropriate information about the statistical significance of the experiments?

Answer: [No]

Justification: The paper does not report error bars, variance, confidence intervals, or significance tests.

Guidelines:

- The answer NA means that the paper does not include experiments.
- The authors should answer "Yes" if the results are accompanied by error bars, confidence intervals, or statistical significance tests, at least for the experiments that support the main claims of the paper.
- The factors of variability that the error bars are capturing should be clearly stated (for example, train/test split, initialization, random drawing of some parameter, or overall run with given experimental conditions).
- The method for calculating the error bars should be explained (closed form formula, call to a library function, bootstrap, etc.)
- The assumptions made should be given (e.g., Normally distributed errors).
- It should be clear whether the error bar is the standard deviation or the standard error of the mean.
- It is OK to report 1-sigma error bars, but one should state it. The authors should preferably report a 2-sigma error bar than state that they have a 96% CI, if the hypothesis of Normality of errors is not verified.

- For asymmetric distributions, the authors should be careful not to show in tables or figures symmetric error bars that would yield results that are out of range (e.g. negative error rates).
- If error bars are reported in tables or plots, The authors should explain in the text how they were calculated and reference the corresponding figures or tables in the text.

8. **Experiments compute resources**

Question: For each experiment, does the paper provide sufficient information on the computer resources (type of compute workers, memory, time of execution) needed to reproduce the experiments?

Answer: [Yes]

Justification: Section 4.1 provides specifics: SFT on 4 A100 GPUs, RL on 16 GH200 GPUs, with training durations, batch sizes, and generation counts.

Guidelines:

- The answer NA means that the paper does not include experiments.
- The paper should indicate the type of compute workers CPU or GPU, internal cluster, or cloud provider, including relevant memory and storage.
- The paper should provide the amount of compute required for each of the individual experimental runs as well as estimate the total compute.
- The paper should disclose whether the full research project required more compute than the experiments reported in the paper (e.g., preliminary or failed experiments that didn't make it into the paper).

9. **Code of ethics**

Question: Does the research conducted in the paper conform, in every respect, with the NeurIPS Code of Ethics https://neurips.cc/public/EthicsGuidelines?

Answer: [Yes]

Justification: The research aligns with NeurIPS ethical guidelines, including transparency of method, release plans, and no direct risks related to human subjects or private data.

Guidelines:

- The answer NA means that the authors have not reviewed the NeurIPS Code of Ethics.
- If the authors answer No, they should explain the special circumstances that require a deviation from the Code of Ethics.
- The authors should make sure to preserve anonymity (e.g., if there is a special consideration due to laws or regulations in their jurisdiction).

10. **Broader impacts**

Question: Does the paper discuss both potential positive societal impacts and negative societal impacts of the work performed?

Answer: [No]

Justification: The paper does not explicitly address societal impacts such as misuse, fairness, or safety. Although scene graph generation has potential societal implications, this aspect falls outside the scope of the current work.

Guidelines:

- The answer NA means that there is no societal impact of the work performed.
- If the authors answer NA or No, they should explain why their work has no societal impact or why the paper does not address societal impact.
- Examples of negative societal impacts include potential malicious or unintended uses (e.g., disinformation, generating fake profiles, surveillance), fairness considerations (e.g., deployment of technologies that could make decisions that unfairly impact specific groups), privacy considerations, and security considerations.
- The conference expects that many papers will be foundational research and not tied to particular applications, let alone deployments. However, if there is a direct path to any negative applications, the authors should point it out. For example, it is legitimate

to point out that an improvement in the quality of generative models could be used to generate deepfakes for disinformation. On the other hand, it is not needed to point out that a generic algorithm for optimizing neural networks could enable people to train models that generate Deepfakes faster.

- The authors should consider possible harms that could arise when the technology is being used as intended and functioning correctly, harms that could arise when the technology is being used as intended but gives incorrect results, and harms following from (intentional or unintentional) misuse of the technology.
- If there are negative societal impacts, the authors could also discuss possible mitigation strategies (e.g., gated release of models, providing defenses in addition to attacks, mechanisms for monitoring misuse, mechanisms to monitor how a system learns from feedback over time, improving the efficiency and accessibility of ML).

11. **Safeguards**

Question: Does the paper describe safeguards that have been put in place for responsible release of data or models that have a high risk for misuse (e.g., pretrained language models, image generators, or scraped datasets)?

Answer: [No]

Justification: Since our models are trained on two public datasets, there is no direct risk of misuse.

Guidelines:

- The answer NA means that the paper poses no such risks.
- Released models that have a high risk for misuse or dual-use should be released with necessary safeguards to allow for controlled use of the model, for example by requiring that users adhere to usage guidelines or restrictions to access the model or implementing safety filters.
- Datasets that have been scraped from the Internet could pose safety risks. The authors should describe how they avoided releasing unsafe images.
- We recognize that providing effective safeguards is challenging, and many papers do not require this, but we encourage authors to take this into account and make a best faith effort.

12. **Licenses for existing assets**

Question: Are the creators or original owners of assets (e.g., code, data, models), used in the paper, properly credited and are the license and terms of use explicitly mentioned and properly respected?

Answer: [Yes]

Justification: Datasets and tools like VG150, PSG, vLLM, and trl are used and properly cited with corresponding references.

Guidelines:

- The answer NA means that the paper does not use existing assets.
- The authors should cite the original paper that produced the code package or dataset.
- The authors should state which version of the asset is used and, if possible, include a URL.
- The name of the license (e.g., CC-BY 4.0) should be included for each asset.
- For scraped data from a particular source (e.g., website), the copyright and terms of service of that source should be provided.
- If assets are released, the license, copyright information, and terms of use in the package should be provided. For popular datasets, `paperswithcode.com/datasets` has curated licenses for some datasets. Their licensing guide can help determine the license of a dataset.
- For existing datasets that are re-packaged, both the original license and the license of the derived asset (if it has changed) should be provided.
- If this information is not available online, the authors are encouraged to reach out to the asset's creators.

13. **New assets**

    Question: Are new assets introduced in the paper well documented and is the documentation provided alongside the assets?

    Answer: [Yes]

    Justification: The paper introduces the R1-SGG model and corresponding reward mechanisms, and promises documentation and release of assets.

    Guidelines:

    - The answer NA means that the paper does not release new assets.
    - Researchers should communicate the details of the dataset/code/model as part of their submissions via structured templates. This includes details about training, license, limitations, etc.
    - The paper should discuss whether and how consent was obtained from people whose asset is used.
    - At submission time, remember to anonymize your assets (if applicable). You can either create an anonymized URL or include an anonymized zip file.

14. **Crowdsourcing and research with human subjects**

    Question: For crowdsourcing experiments and research with human subjects, does the paper include the full text of instructions given to participants and screenshots, if applicable, as well as details about compensation (if any)?

    Answer: [NA]

    Justification: The work does not involve human subjects or crowdsourced data collection.

    Guidelines:

    - The answer NA means that the paper does not involve crowdsourcing nor research with human subjects.
    - Including this information in the supplemental material is fine, but if the main contribution of the paper involves human subjects, then as much detail as possible should be included in the main paper.
    - According to the NeurIPS Code of Ethics, workers involved in data collection, curation, or other labor should be paid at least the minimum wage in the country of the data collector.

15. **Institutional review board (IRB) approvals or equivalent for research with human subjects**

    Question: Does the paper describe potential risks incurred by study participants, whether such risks were disclosed to the subjects, and whether Institutional Review Board (IRB) approvals (or an equivalent approval/review based on the requirements of your country or institution) were obtained?

    Answer: [NA]

    Justification: Not applicable since no human subjects are involved.

    Guidelines:

    - The answer NA means that the paper does not involve crowdsourcing nor research with human subjects.
    - Depending on the country in which research is conducted, IRB approval (or equivalent) may be required for any human subjects research. If you obtained IRB approval, you should clearly state this in the paper.
    - We recognize that the procedures for this may vary significantly between institutions and locations, and we expect authors to adhere to the NeurIPS Code of Ethics and the guidelines for their institution.
    - For initial submissions, do not include any information that would break anonymity (if applicable), such as the institution conducting the review.

16. **Declaration of LLM usage**

Question: Does the paper describe the usage of LLMs if it is an important, original, or non-standard component of the core methods in this research? Note that if the LLM is used only for writing, editing, or formatting purposes and does not impact the core methodology, scientific rigorousness, or originality of the research, declaration is not required.

Answer: [No]

Justification: We only use LLMs for writing assistance and proofreading.

Guidelines:

- The answer NA means that the core method development in this research does not involve LLMs as any important, original, or non-standard components.
- Please refer to our LLM policy (`https://neurips.cc/Conferences/2025/LLM`) for what should or should not be described.

