# OpenReview forum: "Compile Scene Graphs with Reinforcement Learning"
_NeurIPS.cc/2025/Conference — Submitted to NeurIPS 2025_

### Official Review · Reviewer_S4eL · 2025-05-31

**Clarity:** 3
**Significance:** 2
**Originality:** 2
**Rating:** 3
**Confidence:** 4

**Summary:**

This paper proposes reinforcement learning for fine-tuning multimodal large language models for the task of scene graph detection. They evaluate the proposed method on the VG150 and PSG datasets and show improved performance compared to the baselines.

**Questions:**

- Please clarify the exact experimental settings for the results reported in Tab. 1, Tab. 2.
- Please clarify the contributions other than training Qwen2-VL [32] using RL [28].
- How does the method perform on VG150 test set compared to the SOTA.

**Ethical Concerns:**

["NO or VERY MINOR ethics concerns only"]

**Final Justification:**

I thank the reviewers for the rebuttal and response to my questions. Most of my concerns are addressed. However, there are some remaining concerns, such as the high computation cost of the model and the ambiguities in the current version of the paper. The high computation cost is even brought up in the authors' rebuttal, as they mention it is not feasible to run MLLMs on the dataset's test set. I increase my rating to borderline reject, but I remain on the reject side given the remaining limitations.

**Limitations:**

No. There are no discussions on the limitations or negative societal impacts. I suggest the authors discuss the limitations of MLLMs and adapting them for the task of scene graph detection/generation, and potential limitations in ambiguous or over-confident responses.

**Quality:**

2

**Strengths And Weaknesses:**

Strengths:
+ The proposed reinforcement learning method improves over the basic supervised fine-tuning and outperforms the listed baselines.
+ Employing multimodal large language models for the scene graph generation/detection task is interesting and well-motivated.

Weaknesses:
- The evaluation is very limited and does not follow the previous work (mean recall at 20/50/100). The paper also lacks comparison to recent SOTA from 2024, which have comparable or better performance to the proposed method.
- The novelty is limited and incremental (combination of [32,28]). Instead of supervised fine-tuning, the baseline LLM (Qwen2-VL-2b/7b-Instruct) [32] is trained using reinforcement learning [28].
- The results are inconsistent and differ from the results reported in previous work. Specifically, many works evaluate on the VG150 test set, while this work evaluates it on the validation set. This makes it incomparable to many related works. In Tab. 2, on the PSG dataset, the reported results for PSGFormer are on mR@20, which is not mentioned in the paper. Since the other reported results are different from the ones reported in other works, it is not clear whether those values are also on mR@20 or not.
- There is a large gap in the number of parameters between the LLM and non-LLM based methods (Millions vs. Billions). Also the authors do not report the number of parameters for the non-LLM based methods, which have been reported and are available in the original works.
- There are some missing related works that are neither cited nor compared against:
[a] Lorenz, Julian, et al. "A Fair Ranking and New Model for Panoptic Scene Graph Generation." European Conference on Computer Vision. Cham: Springer Nature Switzerland, 2024.
[b] Zhou, Zijian, Miaojing Shi, and Holger Caesar. "VLPrompt: Vision-language prompting for panoptic scene graph generation." arXiv preprint arXiv:2311.16492 (2023).

Minor:
- The authors' response to Q16 on the NeurIPS paper checklist asking about using LLM's is "No" and is said to be only for assistance for writing and proofreading, while LLMs are part of the main methodology.
- The authors mentioned they have discussed the limitations in Sec 4.5. However, there are no discussions on the limitations of the proposed method in this section.

---

> ### Author Rebuttal · Authors · 2025-07-25
>
> We thank Reviewer S4eL for the valuable time and constructive feedback. We address the raised concerns as follows:
>
> 1. **Clarification of Experimental Settings (Tab. 1 & Tab. 2)**:
> All experiments follow the standard SGDET protocol [34,30], which jointly evaluates object detection and relationship prediction without pre‑defined boxes. Traditional task‑specific SGG models output triplets with confidence scores, so **R@K and mR@K (K=20/50/100) are computed by ranking predictions by score and taking the top‑k triplets.
> However, MLLM generates triplets without scores, making top‑k ranking meaningless**. Following prior works using MLLMs for SGG (e.g., LLaVA‑SpaceSGG [35]), we directly report R@20 and mR@20 as the main recall metrics. We will clarify this in the next version.
> 2. **Contributions Beyond “Training Qwen2‑VL with RL”**:
> Our contributions go beyond applying GRPO [28] to Qwen2‑VL [32]:
> 	- Graph‑centric reward design. We introduce three rule‑based reward variants (Hard Recall, Hard Recall+Relax, Soft Recall) that explicitly align with SGG evaluation metrics. These rewards are novel for structured vision tasks and enable fine‑grained node/edge‑level feedback.
> 	- Comprehensive analysis of reward sparsity and cross‑dataset generalization: We systematically study the impact of reward alignment (Tab. 4) and show that RL enhances domain transferability and tail‑class recall (Fig. 3 & 4).
> 	- A unified end‑to‑end MLLM framework (R1‑SGG): To our knowledge, **this is the first work** to integrate visual instruction tuning and reinforcement learning for end‑to‑end SGG with MLLMs, going beyond caption‑based or text‑only LLM methods.
>
> 3. **Comparison with 2024 SOTA**:  We already include results of OvSGTR [ECCV 2024], ASMv2 [ECCV 2024], and LLaVA‑SpaceSGG [WACV 2025] in Tab. 1 and Tab. 2. These methods represent recent SOTA approaches for open‑vocabulary or instruction‑tuned SGG.
>
> 4. **Inconsistent results (why validation set?)**:
> The VG150 test set has ~26k images. VS3 [CVPR 2023] and OvSGTR [ECCV 2024] retain ~15k images that were not used in GLIP pre‑training, but running inference on the remaining 15k images is still extremely expensive for large MLLMs. To reduce computational burden while ensuring fair comparison, we report results on the 5k‑image validation set. Importantly, all models in Tab. 1 are evaluated on the same validation split, which is i.i.d. to the test set, so the comparison remains fair.
> To demonstrate this, we report the performance on the VG150 test set as follows.
> | Split | Model        | Failure Rate| AP@50 | Recall | mRecall |
> |:------|:-------------|------:|------:|-------:|--------:|
> | Test  | IMP          |  - | –   |  17.7  |   2.7   |
> | Test  | MOTIFS       |  -|  –   |  25.5  |   5.0   |
> | Test  | OvSGTR       |-| 36.8  |  27.8  |   5.2   |
> | Test  | R1‑SGG (7B)  |  0.08%| 20.8   |   27.5    |    11.2   |
> | Val.  | IMP          | -|20.9  |  17.9  |   2.7   |
> | Val.  | MOTIFS       | -|29.6  |  27.2  |   7.8   |
> | Val.  | OvSGTR       | -| 33.4  |  26.7  |   5.8   |
> | Val.  | R1‑SGG (7B)  | 0.08%| 19.5  |  23.8  |  11.4   |
>
> For results like PSGFormer in Table 2, we follow previous work LLaVA‑SpaceSGG [WACV 2025] which reports the R@20 as the recall and mR@20 as the mRecall.  The reason is explained in the **Clarification of Experimental Settings (Tab. 1 & Tab. 2)** section.
>
> 5. **Parameter Gap**: It is obvious that MLLMs have more parameters than non-LLM models. However, the goal of R1‑SGG is to study whether RL can enhance structured reasoning in MLLMs, not to directly compete with lightweight SGG detectors.
>
> 6. **Missing Related Works**:  We thank the reviewer for pointing this out. We will cite and discuss VLPrompt (arXiv:2311.16492) and Julian Lorenz et al. (ECCV 2024)
>
> 7. **Discussion of Limitations**: As discussed in To Think or Not Think? (Sec. 4.5), RL alone is not sufficient to trigger abstract reasoning patterns such as CoT. Additional SFT on CoT‑specific data may further enhance reasoning.

---

### Official Review · Reviewer_nLRL · 2025-06-09

**Clarity:** 3
**Significance:** 2
**Originality:** 3
**Rating:** 3
**Confidence:** 4

**Summary:**

This paper introduces an RL framework for the scene graph generation task. The proposed R1-SGG utilizes three graph-centric rewards to align the prediction with the location and relation ground truth. Furthermore, the format consistency reward is proposed to guide the model output to the expected structural schema. The experiments are conducted on two SGG datasets, and the authors compare R1-SGG with specific models, MLLMs, and the SFT baseline.

**Questions:**

- The domain generalization ability is the most important reason for applying RL to SGG from my perspective. However, I'm not convinced by the results in the current manuscript. I recommend that the authors polish this part. For example, it's helpful to understand the RL for SGG by providing an analysis on why the mean recall of R1-SGG is higher than all other baselines.
- I do not understand why the baseline with zero recall or AP, such as GPT-4o, is presented. This either indicates the baseline is not set correctly or the comparison is meaningless.

**Ethical Concerns:**

["NO or VERY MINOR ethics concerns only"]

**Final Justification:**

After reading the discussion and other reviewers' comments, I think the results in the current manuscript do not provide enough support for the authors' claims and the method is limited in novelty. So my score remains the same.

**Limitations:**

I do not find the discussion on limitations even in Section 4.5.

**Quality:**

2

**Strengths And Weaknesses:**

# Strengths
- The research problem is interesting.
- This paper is well-written and easy to follow.

# Weaknesses
- The necessity of using RL for SGG is not convincingly demonstrated by the provided results. First, the AP and recall metrics are lower than those achieved by task-specific models in VG150, which require significantly lower training and inference costs. Second, although the authors present experiments on domain generalization in Table 3, the RL-based approach does not yield meaningful performance under this setting. I find it difficult to see its practical value in this context.
- The format failure rate is not a particularly important issue in the SGG task, and the observation that RL achieves a lower format failure rate compared with SFT is a surprising discovery. It's confusing why this is emphasized by the authors.
- The proposed method appears to be a straightforward application of GRPO to the SGG task, without substantial novelty in its adaptation or design.

---

> ### Author Rebuttal · Authors · 2025-07-25
>
> We thank Reviewer nLRL for the valuable time and constructive feedback. We address the raised concerns as follows:
>
> 1. **Necessity of RL**:
> First, it is reasonable that task‑specific models with dense object detectors achieve higher AP and recall. However, our goal is to develop an effective MLLM for SGG, not to outperform specialized SGG models. Second, SFT cannot directly optimize recall‑based SGG metrics, as all tokens are equally weighted in cross‑entropy loss. Our RL stage introduces graph‑centric rewards aligned with SGDET metrics, yielding large gains (e.g., Recall 5.47% → 21.09%, failure rate 72.42% → 0.10%, Table 1). RL is therefore essential for enabling MLLMs to generate accurate, structurally valid scene graphs in an end‑to‑end way.
> 2. **Format Failure Rate**:
> We report format failure rate not as the main contribution but as a supporting result. MLLMs often fail to follow structured output formats, leading to unusable scene graphs. RL greatly reduces such failures (e.g., 72.42% → 0.10%, Table 1), which is critical for practical deployment of end‑to‑end SGG.
> 3. **Novelty**:
> Beyond using GRPO, we design and study a series of scene‑graph‑specific rewards (Hard Recall, Hard Recall+Relax, Soft Recall, and format consistency). We analyze their sparsity and alignment with SGDET metrics (Table 4). **Our framework is the first to achieve end‑to‑end SGG with an MLLM using RL, demonstrating substantial gains over SFT and zero‑shot MLLMs**.
> 4. **Domain Generalization**:
> Table 3 shows that R1‑SGG‑Zero (RL‑only) generalizes better across datasets (e.g., Recall 13.83% vs. 3.03% for SFT when trained on VG150 and tested on PSG).  Furthermore, Figures 3 and 4 show per‑predicate recall, where R1‑SGG achieves higher recall, especially on tail classes. This explains why R1‑SGG obtains higher mean recall than all baselines—it is more effective at recognizing rare predicates.
> 5. **Baselines like GPT‑4o**:
> Commercial MLLMs are included to illustrate that even strong models fail in zero‑shot SGG, motivating the need for SFT + RL. Their low AP and recall stem from limited grounding ability—we have no control over how they process images or map predicted bounding boxes to correct locations. This highlights the necessity of training an MLLM specifically for SGG.
>
> 6. **Discussion of Limitations**: As discussed in To Think or Not Think? (Sec. 4.5), RL alone is not sufficient to trigger abstract reasoning patterns such as CoT. Additional SFT on CoT‑specific data may further enhance reasoning.

---

> ### Comment · Reviewer_nLRL · 2025-08-03
>
> Thank the authors for the rebuttal. However, there are no additional experiments to support the authors' claims about domain generalization, and the evaluation in the current manuscripts may not be comprehensive enough, which is also mentioned by other reviewers. I would recommend that the authors polish the experiment design to make the claim of this paper more solid.

---

> > ### Author Response · Authors · 2025-08-03
> >
> > Regarding the comment *“there are no additional experiments to support the authors' claims about domain generalization, and the evaluation in the current manuscript may not be comprehensive enough”*, could you please specify which additional experimental settings you would like to see? Our experiments in Table 3 already address this concern, as VG150 and PSG have different domain gaps and category distributions. For example, we trained models on VG150 and directly tested them on PSG. To demonstrate this, we calculate the recall of triplets containing novel objects or relationships in this setting. In the following table, *Base* refers to triplets whose objects and relationships all belong to VG150 categories, whereas *Novel* refers to triplets containing at least one novel object or relationship category.
> >
> > | Model | Recall (Base) | Recall (Novel) |
> > |-------|------|-------|
> > | Qwen2-VL-7B-Instruct     | 3.23%  | 2.20%    |
> > |R1-SGG-Zero  (7B) | 15.77% | 4.42%|
> > These results show that performance improves on both Base and Novel categories.

---

### Official Review · Reviewer_wFe2 · 2025-07-06

**Clarity:** 3
**Significance:** 3
**Originality:** 2
**Rating:** 2
**Confidence:** 4

**Summary:**

The authors present R1-SGG, a framework that turns a multimodal LLM (Qwen2-VL 2 B/7 B) into an end-to-end scene-graph generator. Basically, they firstly do SFT to align with scene-graph JSON output using prompt-response pairs on VG150 and PSG, and then use GRPO to train with the reward, which is the Recall of the triplets and other softer variants

**Questions:**

See weekness.

I think the main issue is (1) performance, MLLM is limited by object detection capability, which drives the SGG performance less than the traditional method, even if with RL. How to fill the gap would be a key issue.

(2) less analysis on the thinking process using RL. The RL can work by reshaping the prob distribution of the text by rewarding, train with direct metric with RL should generally work, but the emerging of CoT would be a positive factor to improve reasoning capability. I would expect to see the CoT with RL.

(3) Also, I would like the author to further illustrate why the scene graph task is still worth studying, since it is an intermediate result, not the finally output. Previous years people research Scene graph as an intermediate layer for visual representation, but now since MLLM has been so powerful, it can directly generate our target output, such as a caption, QA, instead of an intermediate layer of scene graph. And the performance of the scene graph is always limited since this data is not able to scale up naturally, and the abstraction is purely human-defined with object class and relation class, which are over-simplified.

**Ethical Concerns:**

["NO or VERY MINOR ethics concerns only"]

**Final Justification:**

There is a lot of questions can be answered via experiment but the author simply admit the drawbacks or mention it is future work. I think this author is not spending enough time to rebuttal. Therefore I tends to reject this paper.

**Limitations:**

There is no limitation section of the paper.

See weakness section

**Quality:**

2

**Strengths And Weaknesses:**

Advantage:
1.	This paper applies RL to the scene graph task, showing whether RL is able to assist the performance of SGG for MLLM, which has a good motivation.
2.	This paper also ablate the some necessary aspects of RL, such as different reward design, computes/speed balance, sequence length.
3.	The paper is well written and easy to follow.

Disadvantages:
1.	Insufficient evaluation metric. I would like to know how well the model predicts the relation and object classes given two objects’ bounding box locations, and the prediction of the relation classes given the bounding box and object class. Seems MLLM is not convenient for such prompt structure changed evaluation?
2.	In Table 4, why is soft recall better at failure rate than hard recall, but not the case for mRecall?
3.	Does the <think> token really being used? Seems that the model is not using <think> token to enclose the thinking process, and attribute to the 2B/7B LLM is insufficient to follow the language format. But there seems no format reward being designed, and also larger LLM such as Qwen 32B is more capable on format/instruct following, perhaps the author could try that base model.
4.	Seems that the object detection capability (AP50) is still much weaker than the traditional method, as well as recall@50. There are also works that can further enhance the object detection capability of MLLM using GRPO, showing that the detection could be further improved using RL. So I wonder if it is mainly because of the detection capability is improved, or the relation understanding capability is improved.
5.	The reported recall@50 is much lower than related work. Such as Tab.2 R@50 of paper [1].
6.     This paper shows three reward in methods, but I think that should be in ablation section.
[1]  Learning to Generate Language-supervised and Open-vocabulary Scene Graph using Pre-trained Visual-Semantic Space, CVPR2023

---

> ### Author Rebuttal · Authors · 2025-07-25
>
> We thank Reviewer wFe2 for the valuable time and constructive feedback. We address the raised concerns as follows:
> 1. **Insufficient evaluation metric**:
> We follow the standard SGDET protocol, which jointly evaluates object detection and relationship prediction. SGDET is the most challenging SGG setting, as it requires the model to predict object categories, bounding boxes, and relationships simultaneously, unlike PredCls (predicting only relationships given ground-truth boxes and labels) or SGCls (predicting relationships and object labels given ground-truth boxes). **Since our goal is to achieve end-to-end SGG, SGDET is the most suitable evaluation protocol**. The evaluation cases the reviewer mentions (e.g., SGCls) are less practical for MLLMs, as they require explicit prompts with ground‑truth bounding boxes.
>
> 2. **Soft vs. Hard Recall (Tab. 4)**: As shown in Table 4, the failure rates are nearly identical for Hard Recall and Soft Recall (0.08 % vs. 0.06 %). However, the ablation study demonstrates that aligning the reward function with the evaluation metric (Hard Recall) leads to higher mRecall (11.43 % vs. 5.61 %). This confirms that although Soft Recall provides denser reward signals, metric‑aligned rewards are more effective for improving final evaluation performance.
>
> 3. **Format following and \<think\> tokens**: Our method includes a format reward (Sec. 3.3.1) encouraging adherence to the \<think\>...\</think\>\<answer\>...\</answer\> structure. However, Qwen2‑VL‑2B/7B still struggles to produce CoT consistently (Sec. 4.5). We agree that larger models (e.g., Qwen 32B) could further improve format following; we will mention this as a limitation and a promising direction.
>
> 4. **Object‑detection capability and performance gap to traditional SGG**:
> We acknowledge that the current AP@50 is lower than that of detector‑based methods. However, R1‑SGG achieves substantial improvements in AP@50, Recall, and mRecall over both SFT and zero‑shot MLLMs (Tables 1–2) and effectively reduces bias on tail predicates (Figs. 3–4). These gains result from enhancements in both object detection and relation reasoning. For example, with qwen2vl‑2b + SFT, AP@50 increases from 36.05 to 39.28, and Recall rises from 22.06 to 38.49 after RL fine‑tuning.
>
> 5. **Report performance**: Following prior work such as LLaVA‑SpaceSGG [35], we report R/mR@20 as Recall/mRecall for non‑LLM baselines. For MLLMs, however, the models generate unordered triplet sets without confidence scores, making top‑k ranking (R/mR@K) ill‑defined. Therefore, we adopt Recall and mean Recall without top‑k filtering to provide a fair and consistent evaluation across all MLLM settings. We will clarify this in next version.
>
> 6. **Chain‑of‑Thought (CoT) with RL**:
> As discussed in Sec. 4.5, current models seldom produce \<think\> content even with the format reward. We plan to explore CoT‑specific SFT before RL as a potential way to enhance structured reasoning, and we will clarify this as a future direction.
> However, as highlighted in [1], CoT brings the largest benefits in tasks requiring multi‑step reasoning such as math or logic, while the gains are much smaller for other tasks. Since SGG is primarily a perception task (predicting objects, boxes, and relations), the benefits of CoT are expected to be limited compared to multi-step reasoning tasks.
>
> 7. **why the scene graph task is still worth studying**:  Scene graphs remain valuable as structured, compositional representations for tasks such as embodied AI, semantic communication, and medical report generation, etc. Saying that the advent of LLMs makes the task meaningless is like claiming that “because LLMs exist, NLP is dead.” It is unreasonable to assume that a task becomes obsolete simply because we now have powerful LLMs.
>
> 8. **Discussion of Limitations**: As discussed in To Think or Not Think? (Sec. 4.5), RL alone is not sufficient to trigger abstract reasoning patterns such as CoT. Additional SFT on CoT‑specific data may further enhance reasoning.
>
> [1] Sprague, Zayne, et al. "To cot or not to cot? chain-of-thought helps mainly on math and symbolic reasoning." ICLR 2025.

---

> > ### Comment · Reviewer_wFe2 · 2025-08-04
> > **Insufficient Rebuttal**
> >
> > 1. For PredCls, since MLLM is casual, you can force the generated triplet sequence with `<object1 box1, object2 box2` as prompt and expect to see `relation>` as next token prediction, so that enable the evaluation with PreCls. Similar way to evaluate SGCls. You can specify the format in the system prompt.
> > 2. I don't think the author is doing the rebuttal responsibly. I will reject this paper.

---

> > > ### Author Response · Authors · 2025-08-04
> > >
> > > First, as we mentioned, **SGDET is the most challenging setting compared to PredCls and SGCls**. Second, due to architectural differences, **MLLMs cannot extract features corresponding to a specific region or object, nor can they obtain features representing relationships between object pairs—unlike object detectors such as Faster R-CNN or DETR. Therefore, it is not meaningful to evaluate an MLLM’s PredCls or SGCls performance**. Regarding your suggestion—“you can force the generated triplet sequence with <object1 box1, object2 box2> as the prompt”—this is infeasible during testing, as the ground truth is unknown. In real applications, MLLMs have no access to such information.

---

### Official Review · Reviewer_gRpx · 2025-07-08

**Clarity:** 4
**Significance:** 4
**Originality:** 4
**Rating:** 4
**Confidence:** 5

**Summary:**

The paper proposes a novel reinforcement-learning scene graph generation pipeline to achieve end-to-end generation. It proposes a multimodal LLM trained via supervised fine-tuning and a reinforcement learning based refinement to generate scene graphs. The design of graph centric rewards provides an effective way to evaluate the semantic and spatial alignment between predictions and GT for entities and relationships.

**Questions:**

Please refer to the weaknesses.

**Ethical Concerns:**

["NO or VERY MINOR ethics concerns only"]

**Final Justification:**

After reading reviews of other reviewers and authors' responses, I maintain my rating as boardline accept.

**Limitations:**

Yes.

**Quality:**

4

**Strengths And Weaknesses:**

Strengths:
1. A novel scene graph generation pipeline for language model enhanced visual perception. The proposed reinforcement-learning augmented scene graph generation shows strong perception ability against multimodal language models.
2. The designed graph-centric rewards well evaluate the semantic and spatial alignment between predictions and GT for entities and relationships, which has not been explored before.
3. The proposed framework enhances the perception ability for multimodal LLMs in the area of understanding adn reasoning about scene graphs.

Weaknesses:
1. Considering that relationships in scene graphs may not be exactly matched with MLLM-generated predictions, how do the authors tackle this situation?
2. The open-domain ability should be discussed in the experimental analysis.
3. The authors built the benchmark based on the scene graph detection task, which is one of three scene graph generation tasks.  In my opinion, the paper mainly focuses on evaluating the ability of generating object relationships rather than scene graph generation. I hope the authors can offer additional evaluation metric specifically for scene relationship generation of MLLM.

---

> ### Author Rebuttal · Authors · 2025-07-25
>
> We sincerely thank Reviewer gRpx for the constructive feedback and positive evaluation of our work. We address the raised concerns as follows:
>
> Q1:
> In SGG, recall is the primary evaluation metric.
> For triplets that do not exactly match the ground truth, they are naturally excluded from the recall computation and do not contribute to the final score.
> However, during inference our models can still generate semantically correct but unmatched triplets, such as “<woman.13, wearing, pant.8>” shown in Fig. 6. These triplets are plausible yet absent from the ground truth annotations, which highlights that our approach can capture additional valid relationships beyond the labeled dataset.
>
>
>
> Q2: Our method already investigates open-domain capability by evaluating models under different settings in Table 3, including cases without predefined object and relation categories (“w/o cats.”) and cross-dataset generalization (training on VG150 and evaluating on PSG, and vice versa).
> The results demonstrate that R1-SGG-Zero shows stronger cross-domain transfer, while R1-SGG achieves higher in-domain performance when initialized with SFT.
>
>
> Q3:
> We follow the standard SGDET protocol, which jointly evaluates object detection and relationship prediction.
> SGDET is the most challenging SGG setting, as it requires the model to predict object categories, bounding boxes, and relationships simultaneously, unlike PredCls (predicting only relationships given ground-truth boxes and labels) or SGCls (predicting relationships and object labels given ground-truth boxes).
> Since our goal is to achieve end-to-end SGG, SGDET is the most suitable evaluation protocol.

---

> ### Comment · Area_Chair_Pvfw · 2025-08-08
>
> Dear Reviewer gRpx,
>
> This is the last call for author-reviewer discussion, which will ends on Aug 8. Could you please read authors' rebuttal and confirm whether your concerns have been addressed asap? Thank you.
>
> Best,
>
> AC

---

### Decision · Program_Chairs · 2025-09-17

**Decision:**

Reject

**Comment:**

This work focused on the task of scene graph generation (SGG) with multimodal LLM. The key challenge is to adapt token-level next-word prediction of LLM to structured graph generation in an end-to-end manner. To tackle this challenge, this work proposed R1-SGG to incorporate reinforcement learning with SGG using graph-centric rewards to capture semantic and spatial alignment at object-level and relation-level beyond token-level. Experiments are conducted on VG150 and PSG benchmarks to validate the effectiveness of R1-SGG.

The main strengths are that (1) applying RL to SGG with MLLMs is novel, (2) reward design for semantic and spatial alignment are meaningful, (3) the overall writing quality is good. The main weaknesses are (1) the evaluation setting is not standard and lack of some key metrics, (2) the performance gain is limited, (3) the application of GRPO is straightforward concerning the novelty.

After rebuttal and discussions, reviewers have remaining concerns on (1) insufficient evaluation metrics and (2) missing experiments to support claims about domain generalization. Although the idea is novel to SGG with MLLM, the tech novelty of implementation is not strong, and the evaluation have several major concerns to be resolved. Therefore, AC does not recommend accepting this paper.